# DO-EM: Density Operator Expectation Maximization

## Abstract

Density operators, quantum generalizations of probability distributions, are gaining prominence in machine learning due to their foundational role in quantum computing. Generative modeling based on density operator models (**DOMs**) is an emerging field, but existing training algorithms – such as those for the Quantum Boltzmann Machine – do not scale to real-world data, such as the MNIST dataset. The Expectation-Maximization algorithm has played a fundamental role in enabling scalable training of probabilistic latent variable models on real-world datasets. *In this paper, we develop an Expectation-Maximization framework to learn latent variable models defined through **DOMs** on classical hardware, with resources comparable to those used for probabilistic models, while scaling to real-world data.* However, designing such an algorithm is nontrivial due to the absence of a well-defined quantum analogue to conditional probability, which complicates the Expectation step. To overcome this, we reformulate the Expectation step as a quantum information projection (QIP) problem and show that the Petz Recovery Map provides a solution under sufficient conditions. Using this formulation, we introduce the Density Operator Expectation Maximization (DO-EM) algorithm – an iterative Minorant-Maximization procedure that optimizes a quantum evidence lower bound. We show that the **DO-EM** algorithm ensures non-decreasing log-likelihood across iterations for a broad class of models. Finally, we present Quantum Interleaved Deep Boltzmann Machines (**QiDBMs**), a **DOM** that can be trained with the same resources as a DBM. When trained with **DO-EM** under Contrastive Divergence, a **QiDBM** outperforms larger classical DBMs in image generation on the MNIST dataset, achieving a 40–60% reduction in the Fréchet Inception Distance.

## 1 Introduction

Recent advances in quantum hardware and hybrid quantum-classical algorithms have fueled a surge of interest in developing learning models that can operate effectively in quantum regimes [1]. Classical models rely on probability distributions; quantum systems generalize these to density operators - positive semi-definite, unit-trace operators on Hilbert spaces—that encode both classical uncertainty and quantum coherence [2]. While there is considerable progress made in quantum supervised learning, there is relatively less progress in unsuperviced learning [3].

Latent variable models (LVMs) are a cornerstone of unsupervised learning, offering a principled approach to modeling complex data distributions through the introduction of unobserved or hidden variables [4]. These models facilitate the discovery of underlying structure in data and serve as the foundation for a wide range of tasks, including generative modeling, clustering, and dimensionality reduction. Classical examples such as Gaussian Mixture Models, Factor Analysis, and Hidden Markov Models [5, 6] exemplify the power of latent variable frameworks in capturing dependencies and variability in observed data. In recent years, LVMs have formed the conceptual backbone of

deep generative models including Variational Autoencoders [7], Generative Adversarial Networks [8], and Diffusion-based models [9]. The EM algorithm [10, 11] has been instrumental in deriving procedures for learning latent variables models. These algorithms are often preferred over algorithms which directly maximizes likelihood.

The study of Density Operator-based Latent Variable Models (**DO-LVM**) remains in its early stages, with foundational questions around expressivity, inference, and learning still largely unexplored [12–14]. Leveraging the modeling power of **DO-LVMs** on real-world data remains a significant challenge. Existing approaches rarely scale beyond 12 visible units—limited by restricted access to quantum hardware, the exponential cost of simulating quantum systems, and the memory bottlenecks associated with representing and optimizing **DO-LVMs** on classical devices. As a result, it is currently infeasible to empirically assess whether **DO-LVMs** offer any practical advantage on real-world datasets in terms of modeling power. EM based algorithms can provide a simpler alternative to existing learning algorithms for **DO-LVMs** which directly maximizes the likelihood. However deriving such algorithms in Density operator theoretic setup is extremely challenging for a variety of reasons, Most notably there are operator theoretic inequalities, such as Jensen Inequality, which can be directly applied to derive an Evidence lower bound(ELBO) style bound for **DO-LVMs**. Precise characterization of models which are compatible with such bounds and their computational behaviour remains an important area of investigation. In this paper we bridge these research gaps by making the following contributions.

- A Density Operator Expectation-Maximization (**DO-EM**) algorithm is specified using Quantum Information Projection in Algorithm 1. **DO-EM** guarantees log-likelihood ascent in Theorem 4.4 under mild assumptions that retain a rich class of models.

- A Quantum Evidence Lower Bound (`QELBO`) for the log-likelihood is derived in Lemma 4.1 from a minorant-maximization perspective leveraging the Monotonicity of Relative Entropy.

- **DO-LVMs** are specialized to train on classical data in Section 5 using the **DO-EM** algorithm. This specialization we call **CQ-LVMs**, a class of models with quantum latent variables, can train real world data due to a decomposition proved in Theorem 5.1.

- Quantum-interleaved deep Boltzmann machines (`QiDBM`), a quantum analog of the DBM is defined in Section 5.1. The well known Contrastive Divergence (CD) algorithm for Boltzmann machines is adapted to the `QiDBM`, which when used with **DO-EM** algorithm in Section 5.1, allows `QiDBM`s to be trained on MNIST-scale data.

- First empirical evidence of a modeling advantage when training **DO-LVMs** on standard computers with real-world data is provided in Section 6. `QiDBM`s trained using CD on the MNIST dataset achieve a 40–60% lower Fréchet Inception Distance compared to state-of-the-art deep Boltzmann machines.

## 2 Preliminaries

**Notation**  The $\ell^2$-norm of a column vector $\mathbf{v}$ in a Hilbert space $\mathcal{H}$ is given by $||\mathbf{v}||_2 = \sqrt{\mathbf{v}^\dagger \mathbf{v}}$ where $\mathbf{v}^\dagger$ denotes the conjugate transpose of $\mathbf{v}$. The set of Hermitian (self-adjoint) operators $\mathcal{O} = \mathcal{O}^\dagger$ on $\mathcal{H}$ is denoted by $\mathfrak{L}(\mathcal{H})$. The positive-definite subset of $\mathfrak{L}(\mathcal{H})$ is denoted by $\mathfrak{L}_+(\mathcal{H})$. The Kronecker product between two operators is denoted $A \otimes B$ and their direct sum is denoted $A \oplus B$ [15]. The identity operator on $\mathcal{H}$ is denoted $I_\mathcal{H}$. The null space of an operator $A \in \mathcal{H}$ is denoted by $\ker(A)$.

**Latent variable models and EM algorithm**  Latent Variable Models (LVMs) [4] specify the probability distribution of random variables $V = [V_1, \ldots, V_{d_V}]$ through a joint probability model

$$P(V = \mathrm{v} \mid \theta) = \sum_h P(V = \mathrm{v}, H = \mathrm{h} \mid \theta)$$

where $H = [H_1, \ldots, H_{d_L}]$ are unobserved random variables. Learning an LVM from data, a problem of great interest in Unsupervised Learning [5], refers to estimating the model parameters $\theta$ from a dataset $\mathcal{D} = \{\mathrm{v}^{(1)}, \ldots, \mathrm{v}^{(N)}\}$ consisting of i.i.d instances drawn from the LVM. Maximum likelihood-based methods aim to maximize $\mathcal{L}(\theta) = \frac{1}{N} \sum_{i=1}^N \ell_i(\theta)$ where $\ell_i(\theta) = \log P(V = \mathrm{v}^{(i)} \mid \theta)$. The maximization problem is not only intractable in most cases but even gradient-based algorithms, which

can only discover local optima, are difficult to implement because of unwieldy computations in $\ell_i(\theta)$. The EM algorithm [10, 11] is an alternative iterative algorithm with the scheme

$$\theta^{(k+1)} = \underset{\theta}{\operatorname{argmax}} \frac{1}{N} \sum_{i=1}^{N} Q_i(\theta \mid \theta^{(k)}), \text{ where } \ell_i(\theta) \geq Q_i(\theta|\theta^{(k)}) \text{ and } \ell_i(\theta^{(k)}) = Q_i(\theta^{(k)}|\theta^{(k)}).$$

**Boltzmann machines** Boltzmann Machines (BM) are stochastic neural networks that define a probability distribution over binary vectors based on the Ising model in statistical physics [16]. Due to the intractability of learning in fully connected BMs, the Restricted Boltzmann Machine (RBM) was introduced with no intra-layer connections, enabling efficient Gibbs sampling [17–19]. Deep Boltzmann Machines (DBM) [20] stacks RBMs uisng undirected connections and allow for joint training of all layers. The joint probability of a DBM with $L$ layers, $P(\mathbf{v}, \mathbf{h}^1, \ldots, \mathbf{h}^L)$ is defined as

$$P(\mathbf{v}, \mathbf{h}_1, \ldots, \mathbf{h}_{d_L}) = \frac{1}{Z} e^{-E(\mathbf{v}, \mathbf{h}_1, \ldots, \mathbf{h}_{d_L})} \tag{DBM}$$

where $E(\mathbf{v}, \mathbf{h}^1, \ldots, \mathbf{h}^L)$ is called the *Energy Function*, and $Z = \sum_{\mathbf{v}, \mathbf{h}} e^{-E(\mathbf{v}, \mathbf{h}^1, \ldots, \mathbf{h}^L)}$ is the *Partition Function* which is typically intractable to compute. Learning in DBMs is difficult due to intractable posterior dependencies. DBMs are usually trained using variants of the Contrastive Divergence algorithm [18, 21, 22]. A detailed discussion on Boltzmann machines and the Contrastive Divergence algorithm is provided in the Appendix A.

## 2.1 Density operators

A density operator on a Hilbert space $\mathcal{H}$ is a Hermitian, positive semi-definite operator with unit trace [2, 23]. The set of Density operators will be denoted by $\mathcal{P}(\mathcal{H})$, and can be regarded as generalizations of probability distributions. A joint density operator $\rho \in \mathcal{P}(\mathcal{H}_A \otimes \mathcal{H}_B)$ can be *marginalized* to $\rho_A \in \mathcal{P}(\mathcal{H}_A)$ by the partial trace operation $\rho_A = \operatorname{Tr}_B(\rho) = \sum_{i=1}^{d_B} (I_A \otimes \mathbf{x}_i^\dagger) \rho (I_A \otimes \mathbf{x}_i)$ where $\{\mathbf{x}_i\}_{i=1}^{d_B}$ is an orthonormal basis of $\mathcal{H}_B$. Such a $\rho$ is *separable* if it is a convex combination of *product states* $\rho_A \otimes \rho_B$ with $\rho_A \in \mathcal{P}(\mathcal{H}_A)$ and $\rho_B \in \mathcal{P}(\mathcal{H}_B)$.

**Definition 2.1** (Umegaki [24] Relative Entropy). Let $\omega$ and $\rho$ be density operators in $\mathcal{P}(\mathcal{H}_A \otimes \mathcal{H}_B)$ with $\ker(\rho) \subseteq \ker(\omega)$. Their relative entropy is given by $\operatorname{D_U}(\omega, \rho) = \operatorname{Tr}(\omega \log \omega) - \operatorname{Tr}(\omega \log \rho)$.

Lindblad [25] showed that the relative entropy does not increase under the action of the parital trace.

**Theorem 2.2** (Monotonicity of Relative Entropy). *For density operators $\omega$ and $\rho$ in $\mathcal{P}(\mathcal{H}_A \otimes \mathcal{H}_B)$ such that $\ker(\omega) \subset \ker(\rho)$, $\operatorname{D_U}(\omega, \rho) \geq \operatorname{D_U}(\operatorname{Tr}_B\omega, \operatorname{Tr}_B\rho)$.*

Petz [26, 27] showed that Theorem 2.2 is saturated if and only if the Petz Recovery Map reverses the partial trace operation.

**Definition 2.3** (Petz Recovery Map). For a density operator $\rho$ in $\mathcal{P}(\mathcal{H}_A \otimes \mathcal{H}_B)$, the Petz Recovery Map *for the partial trace* $\mathcal{R}_\rho : \mathcal{H}_A \to \mathcal{H}_A \otimes \mathcal{H}_B$ is the map

$$\mathcal{R}_\rho(\omega) = \rho^{1/2} \left( \left( \rho_A^{-1/2} \omega \rho_A^{-1/2} \right) \otimes I_B \right) \rho^{1/2}. \tag{PRM}$$

**Theorem 2.4** (Ruskai's condition). *For density operators $\omega$ and $\rho$ in $\mathcal{P}(\mathcal{H}_A \otimes \mathcal{H}_B)$ such that $\ker(\omega) \subset \ker(\rho)$, $\operatorname{D_U}(\operatorname{Tr}_B\omega, \operatorname{Tr}_B\rho) = \operatorname{D_U}(\omega, \rho)$ if and only if $\log \omega - \log \rho = (\operatorname{Tr}_B\omega - \operatorname{Tr}_B\rho) \otimes I_B$.*

Ruskai's condition can be interpreted as $\omega$ and $\rho$ having the same Conditional Amplitude Operator.

**Definition 2.5** (Conditional Amplitude Operator[28]). The conditional amplitude operator of a density operator $\rho$ in $\mathcal{P}(\mathcal{H}_A \otimes \mathcal{H}_B)$ with respect to $\mathcal{H}_A$ is $\rho_{B|A} = \exp(\log \rho - \log \rho_A \otimes I_B)$.

A detailed discussion on density operators and quantum channels is provided in Appendix B.

## 3 Density operator latent variable models

In this section, we introduce Density Operator Latent Variable Models (**DO-LVM**) and recover existing models such as the Quantum Boltzmann Machine (QBM) as special cases. We discuss the computational challenges of learning such models from observations.

**Definition 3.1** (**DO-LVM** and the Learning Problem). A Density Operator Latent Variable Model (**DO-LVM**) specifies the density operator $\rho_{\mathrm{V}} \in \mathcal{P}(\mathcal{H}_{\mathrm{V}})$ on observables in $\mathcal{H}_{\mathrm{V}}$ through a joint density operator $\rho_{\mathrm{VL}} \in \mathcal{P}(\mathcal{H}_{\mathrm{V}} \otimes \mathcal{H}_{\mathrm{L}})$ as $\rho_{\mathrm{V}} = \mathrm{Tr}_{\mathrm{L}}\left(\rho_{\mathrm{VL}}(\theta)\right)$ where the space $\mathcal{H}_{\mathrm{L}}$ is not observed. Learning a **DO-LVM** is the estimation of model parameters $\theta$ when a target density operator $\eta_{\mathrm{V}} \in \mathcal{P}(\mathcal{H}_{\mathrm{V}})$ is specified. This can be achieved by maximizing the log-likelihood

$$\mathcal{L}(\theta) = \mathrm{Tr}\left(\eta_{\mathrm{V}} \log \rho_{\mathrm{V}}(\theta)\right). \tag{LP}$$

*Remark* 3.2. Maximizing the log-likelihood of a **DO-LVM** is equivalent to minimzing $\mathrm{D}_{\mathrm{U}}(\eta_{\mathrm{V}}, \rho_{\mathrm{V}}(\theta))$.

We specialize **DO-LVMs** to classical datasets in Section 5.

**Hamiltonian-based models** The Hamiltonian is a Hermitian operator $\mathrm{H} \in \mathfrak{L}(\mathcal{H})$ representing the total energy and generalizes the notion of an energy function in classical energy-based models. The model is defined using Gibbs state density matrix analogous to the Boltzmann distribution: $\rho(\theta) = \frac{\exp(\mathrm{H}(\theta))}{Z(\theta)}$ with $Z(\theta) = \mathrm{Tr}\exp(\mathrm{H}(\theta))$ and $\mathrm{H}(\theta) = \sum_r \theta_r \mathrm{H}_r$, where $\mathrm{H}_r \in \mathfrak{L}(\mathcal{H})$ are Hermitian operators and $\theta_r \in \mathbb{R}$ are model parameters. The Quantum Boltzmann Machine is a Hamiltonian-based model inspired by the transverse field Ising model [12]. In this paper, QBM$_{\mathrm{m,n}}$ denotes a model with $m$ visible and $n$ hidden units with

$$\mathrm{H}(\theta) = -\sum_{i=1}^{m+n} b_i \sigma_i^z - \sum_{i>j} w_{ij} \sigma_i^z \sigma_j^z - \sum_{i=1}^{m+n} \Gamma_i \sigma_i^x \tag{QBM}$$

where $\sigma_i^z$ and $\sigma_i^x$ are $2^{m+n} \times 2^{m+n}$ Pauli matrices defined by $\sigma_i^k = \otimes_{j=1}^{i-1} \mathrm{I} \otimes \sigma^k \otimes_{j=i+1}^{m+n} \mathrm{I}$ where $k \in \{x, z\}$, $\sigma^z = \left(\begin{smallmatrix} 1 & 0 \\ 0 & -1 \end{smallmatrix}\right)$, and $\sigma^x = \left(\begin{smallmatrix} 0 & 1 \\ 1 & 0 \end{smallmatrix}\right)$. A QBM is hence a **DO-LVM** with $\rho_{\mathrm{V}}(\theta) = \frac{1}{Z(\theta)} \mathrm{Tr}_{\mathrm{L}} \exp(\mathrm{H}(\theta))$.

Setting $\Gamma_i = 0$ recovers the Boltzmann Machine (BM) [12]. However, the density operator representation of these classical models are plagued by their $2^{m+n} \times 2^{m+n}$ dimensionality. The memory requirements for storing and updating models represented by density operators have been prohibitive for QBMs to scale beyond about 12 visible units.

**Need for an EM algorithm.** As probabilistic LVMs are a special case of **DO-LVMs**, the training challenges they face persist in **DO-LVMs**, which also introduce new operator-theoretic difficulties. Maximizing the log-likelihood of a **DO-LVM** involves operators that do not commute [13]. The direct computation of gradient in Equation (LP) is significantly complicated by the partial trace [29]. Due to the difficulty of working with hidden units, recent work on QBMs have focused on models without hidden units [30, 14, 31, 32]. Demidik et al. [33] studied a Restricted QBM with 12 visible units and 90 hidden units, the largest model studied in literature so far. Refer Appendix B for a detailed survey on QBM literature. Hence, training a QBM, the most popular **DO-LVM** in literature, on real-world data *remains an open challenge*.

Intractability of the gradient of the log-likelihood in probabilistic LVMs is addressed by the EM algorithm. Classical derivations of the EM algorithm fail with density operators since there is no well-defined way to construct conditional density operators [23]. An EM algorithm for density operators using Conditional Amplitude Operators (CAO) was conjectured in Warmuth and Kuzmin [34]. This is insufficient since the CAO does not provide a density operator [28]. In the next section, we appeal to well-known results in quantum information theory to derive an ELBO and EM algorithm for density operators.

## 4 The DO-EM framework

In this section, we develop an algorithmic framework applicable for learning **DO-LVMs** using a density operator expectation maximization framework.

The classical ELBO is derived for each datapoint using conditional probability and Jensen's inequality. This approach fails for density operators due to the absence well-defined quantum conditional probability [23]. In order to derive an ELBO for **DO-LVMs**, we resort to an approach inspired by the chain rule of KL-divergence [35].

**Lemma 4.1** (Quantum ELBO). *Let* $\mathcal{J}(\eta_{\mathrm{V}}) = \{\eta \mid \eta \in \mathcal{P}(\mathcal{H}_{\mathrm{V}} \otimes \mathcal{H}_{\mathrm{L}}) \ \& \ \mathrm{Tr}_{\mathrm{L}}\eta = \eta_{\mathrm{V}}\}$ *be the set of feasible extensions for a target* $\eta_{\mathrm{V}} \in \mathcal{P}(\mathcal{H}_{\mathrm{V}})$. *Then for a **DO-LVM*** $\rho(\theta)$ *and* $\eta \in \mathcal{J}(\eta_{\mathrm{V}})$,

$$\mathcal{L}(\theta) \geq \mathrm{QELBO}(\eta, \theta) = \mathrm{Tr}(\eta \log \rho(\theta)) + S(\eta) - S(\eta_{\mathrm{V}}). \tag{QELBO}$$

171 *Proof sketch:* We provide a proof due to Theorem 2.2 in Appendix C.

172 The classical EM algorithm is a consequence of the ELBO being a minorant of the log-likelihood
173 [36, 37]. However, it is well known that Theorem 2.2 is often not saturated [38–42]. Inspired by an
174 information geometric interpretation of the EM algorithm [43], we study an instance of a quantum
175 information projection problem to saturate QELBO.

## 4.1 A quantum information projection problem

177 In this subsection we study the $I$-projection [35] problem for density operators and show conditions
178 when (PRM) can solve this problem. The problem of Quantum Information Projection (QIP) is stated
179 as follows. Consider a density operator $\omega$ in $\mathcal{P}(\mathcal{H}_A)$ and a density operator $\rho$ in $\mathcal{P}(\mathcal{H}_A \otimes \mathcal{H}_B)$, find
180 $\xi^*$ in $\mathcal{P}(\mathcal{H}_A \otimes \mathcal{H}_B)$ such that

$$\xi^* = \underset{\mathrm{Tr}_B(\xi)=\omega}{\operatorname{argmin}} \, \mathrm{D_U}(\xi, \rho). \tag{QIP}$$

181 To the best of our knowledge, this problem has not been studied in literature. We know from
182 Theorem 2.2 that the theoretical minimum attained by the objective function in QIP is $\mathrm{D_U}(\omega, \mathrm{Tr}_B\rho)$
183 though it is not always saturated. Inspired by this connection, we explore sufficiency conditions for
184 when PRM solves QIP.

185 **Definition 4.2** (Condition S). Two density operators $\omega$ in $\mathcal{P}(\mathcal{H}_A)$ and $\rho$ in $\mathcal{P}(\mathcal{H}_A \otimes \mathcal{H}_B)$ satisfy the
186 sufficiency condition if $\rho$ is full rank, separable, and $[\omega, \mathrm{Tr}_B(\rho)] = 0$.

187 **Theorem 4.3.** *Suppose two density operators $\omega$ in $\mathcal{P}(\mathcal{H}_A)$ and $\rho$ in $\mathcal{P}(\mathcal{H}_A \otimes \mathcal{H}_B)$ such that **Condition***
188 ***S** is satisfied, the solution to the information projection problem* QIP *is* PRM.

189 *Proof sketch:* The statement holds due to the fact that $[\rho, \mathcal{R}_\rho(\omega)] = 0$ under the conditions in the
190 theorem. Thus, $\rho$ and $\mathcal{R}_\rho(\omega)$ obey Ruskai's condition. A detailed proof is provided Appendix C.

## 4.2 DO-EM through the lens of Minorant Maximization

192 In this section, we present the **D**ensity **O**perator **E**xpectation **M**aximization (**DO-EM**) algorithm
193 from a Minorant-Maximization perspective and discuss its advantages over direct maximization of
194 the log-likelihood. We prove that the **DO-EM** algorithm can achieve log-likelihood ascent at every
195 iteration under **Condition S**.

196 For a fixed $\theta^{(\mathrm{old})}$, the QELBO is maximized
197 when $\eta$ is the QIP of $\rho(\theta)$ onto the set of fea-
198 sible extensions. This allows us to define a
199 potential minorant $\mathcal{Q}$ for the log-likelihood.

$$\eta(\theta^{(\mathrm{old})}) = \underset{\mathrm{Tr_L}\eta=\eta_V}{\operatorname{argmin}} \, \mathrm{D_U}(\eta, \rho(\theta^{(\mathrm{old})}))$$

$$\mathcal{Q}(\theta; \theta^{(\mathrm{old})}) = \mathrm{QELBO}(\eta(\theta^{(\mathrm{old})}), \rho(\theta))$$

---

**Algorithm 1 DO-EM**

---

1: **Input:** Target density operator $\eta_V$ and $\theta^{(0)}$
2: **while** not converged **do**
3:    **E Step:** $\eta^{(t)} = \underset{\eta : \mathrm{Tr_L}\eta=\eta_V}{\operatorname{argmin}} \, \mathrm{D_U}(\eta, \rho(\theta^{(t)}))$
4:    **M Step:** $\theta^{(t+1)} = \underset{\theta}{\operatorname{argmax}} \, \mathrm{Tr}(\eta^{(t)} \log \rho(\theta))$

---

200 We use $\mathcal{Q}$ to define the **DO-EM** algorithm in Algorithm 1. Models and QIPs that obey Ruskai's
201 condition provably achieve log-likelihood ascent under the **DO-EM** procedure.

202 **Theorem 4.4** ($\mathcal{Q}$ is a minorant). *Let $\eta_V$ be a target density matrix and $\rho(\theta)$ be a **DO-LVM** trained*
203 *by the **DO-EM** algorithm. If $\rho(\theta^{(t)})$ and its* QIP *onto the set of feasible extensions, $\eta^{(t)}$, obey*
204 *Ruskai's condition, then $\mathcal{Q}$ is a minorant of the log-likelihood. Then, $\mathcal{L}(\theta^{(t+1)}) \geq \mathcal{L}(\theta^{(t)})$, where*
205 $\theta^{(t+1)} = \operatorname{argmax}_\theta \mathcal{Q}(\theta; \theta^{(t)})$.

206 *Proof sketch:* Proof using the saturation of Theorem 2.2 is in Appendix C.

207 **Corollary 4.5.** *For a target density operator $\eta_V$ and model $\rho(\theta)$ satisfying **Condition S**, the E step is*
208 *the Petz recovery map $\mathcal{R}_\rho(\eta_V)$. Moreover, such a model trained using the **DO-EM** algorithm achieves*
209 *provable likelihood ascent at every iteration.*

210 *Proof sketch:* The proof due to Theorem 4.3 and Theorem 4.4 is given in Appendix C.

211 The **DO-EM** algorithm can be considered a density operator analog of the classical EM algorithm.
212 We recover the classical EM algorithm from **DO-EM** for discrete models if $\eta_V$ and $\rho(\theta)$ are diagonal.

213 The **E Step** in **DO-EM** finds a feasible extension $\eta$ whose Conditional Amplitude Operator (`CAO`)
214 is equal to that of the model $\rho(\theta)$. The `PRM` under **Condition S** is the `CAO` reweighted by $\eta_V$ to give
215 a valid density operator. This reduces to classical E step when the `CAO` reduces to the conditional
216 probability and `PRM` reduces to Bayes rule. If the model $\rho$ is of the form $\rho_V \otimes \rho_L$, we recover the
217 conjecture in [34].

218 A log-likelihood involving a partial trace is often intractable. The **M Step** in **DO-EM** algorithm
219 maximizes an expression without the partial trace. The log-likelihood of such expressions may have
220 closed-form expressions for the gradients, for example, using the Lee-Trotter-Suzuki formula [14].
221 In the classical case, this is equivalent to the EM algorithm maximizing a sum of logarithms instead
222 of a logarithm of sums.

223 **Corollary 4.6.** *For a Hamiltonian-based model with E step solution $\eta^{(t)}$, the M step reduces to*

$$\theta^{(t+1)} = \mathrm{argmax}_\theta \, \mathrm{Tr}(\eta^{(t)} \mathrm{H}(\theta)) - \log Z(\theta)$$

224 *Proof sketch:* The proof due to properties of the matrix logarithm is given in Appendix C.

225 However, the memory footprint of **DO-LVM**s remain, preventing the application of these models
226 on real-world data. We specialize **DO-LVM**s and **DO-EM** to train on classical data and achieve
227 practical scale.

## 5 DO-EM for classical data

229 In this section, we specialize **DO-LVMs** and the **DO-EM** algorithm to classical datasets. We
230 assume, for ease of presentation, that the data $\mathcal{D} = \{v^{(1)}, \ldots, v^{(N)}\}$ is sampled from the set $\mathcal{B} =$
231 $\{+1, -1\}^{d_V}$. We consider a $2^{d_V}$-dimensional Hilbert space $\mathcal{H}_V$ with standard basis $\mathfrak{B} = \{\mathbf{v}_i\}_{i=1}^{2^{d_V}}$.
232 There is a one-to-one mapping between elements of $\mathcal{B}$ and $\mathfrak{B}$. For any dataset $\mathcal{D}$, there is an
233 equivalent dataset on $\mathcal{H}_V$ given by $\mathfrak{D} = \{\mathbf{v}^{(1)}, \ldots, \mathbf{v}^{(N)}\}$. The target density operator is then
234 $\eta_V = \frac{1}{N} \sum_{i=1}^{N} \mathbf{v}_i \mathbf{v}_i^\dagger$. A **DO-LVM** on $d_V$-dimensional binary data is therefore a $2^{d_V + d_L} \times 2^{d_V + d_L}$
235 matrix while the target $\eta_V$ is a $2^{d_V} \times 2^{d_V}$ matrix.

236 Specializing **Condition S** to diagonal target density operators, allows the decomposition of a **DO-**
237 **LVM** into direct sums of smaller subspaces, making the **DO-EM** algorithm computationally easier.

238 **Theorem 5.1.** *If $\rho_V$ is diagonal, $\rho$ is separable if and only if $\rho = \oplus_i \rho_L(i)$ and $P(\mathbf{v}_i) = \mathrm{Tr}(\rho_L(i))$*
239 *with $\mathbf{v}_i \in \mathfrak{B}$. The density operator for $\mathcal{H}_L$ for a particular $\mathbf{v}_i$ is then given by $\frac{1}{P(\mathbf{v}_i)} \rho_L(i)$.*

240 *Proof sketch:* See Appendix C.

241 We call models that obey Theorem 5.1 as **CQ-LVMs** since it implies a classical visible probability
242 distribution with a quantum hidden space. `QELBO` can be specialized to each data point for **CQ-LVMs**.

243 **Lemma 5.2.** *For diagonal $\eta_V$ in $\mathcal{P}(\mathcal{H}_V)$, a **DO-LVM** $\rho(\theta)$ satisfies **Condition S** if and only if it*
244 *is of the form in Theorem 5.1. The log-likelihood of these models can then expressed as $\mathcal{L}(\theta) =$*
245 $\frac{1}{N} \sum_{i=1}^{N} \ell_i(\theta)$ *where $\ell_i(\theta) = \log P(\mathbf{v}^{(i)} \mid \theta)$.*

246 *Proof sketch:* The proof is an application of Theorem 5.1 and is given in Appendix C.

247 The decomposition of the log-likelihood into terms for each datapoint, allows the training of models
248 on real-world data since the target densit operator $\eta_V$ does not have to be initialized. We now show
249 that **CQ-LVMs** are a broad class of models that include several Hamiltonian-based models.

250 **Corollary 5.3.** *A Hamiltonian-based model $\rho(\theta) = e^{\mathrm{H}(\theta)}/Z(\theta)$ with $\mathrm{H}(\theta) = \sum_r \theta_r \mathrm{H}_r$ is a **CQ-***
251 ***LVMs** if and only if $\mathrm{H} = \oplus_i \mathrm{H}_i$ where $\mathrm{H}_i$ are Hermitian operators in $\mathfrak{L}(\mathcal{H}_L)$ and $i \in [2^{d_V}]$.*

252 *Proof sketch:* The proof, due to the properties block diagonal matrices, is given in Appendix C. We
253 now specialize `QELBO` and Algorithm 1 to **CQ-LVMs**.

254 **Lemma 5.4.** *For diagonal $\eta_V$ in $\mathcal{P}(\mathcal{H}_V)$ and a **CQ-LVM** $\rho(\theta)$, the log-likelihood of a data point*
255 $\mathbf{v}^{(i)} \in \mathfrak{D}$, $\ell_i(\theta)$ *is lower bounded by*

$$\ell_i(\theta) \geq \mathrm{Tr}\left(\eta_L \log(P(\mathbf{v}^{(i)}|\theta) \rho_L^{(i)}(\theta))\right) - \mathrm{Tr}(\sigma_L \log \sigma_L)$$

256 *for any density operator $\eta_{\mathrm{L}}$ in $\mathcal{P}(\mathcal{H}_{\mathrm{L}})$ with equality if and only if $\eta_{\mathrm{L}} = \rho_{\mathrm{L}}^{(i)}(\theta)$. Hence, the* PRM *is*
257 *given by $\mathcal{R}_{\rho}(\eta_{\mathrm{V}}) = \oplus_i P_{\mathcal{D}}(V = \mathrm{v}_i)\rho_{\mathrm{L}}(i \mid \theta)$.*

258 *Proof sketch:* Application of Lemma 5.4 to Lemma 4.1. Proof is given in Appendix C.

259 This allows us to specialize Algorithm 1 to
260 Algorithm 2, enabling the implementation of
261 **DO-EM** without being restricted by the dimen-
262 sion of $\eta_{\mathrm{V}}$. However, models such as the QBM
263 remain intractable for real-world data due to
264 the normalization term, a problem that exists
265 in classical Boltzmann machines as well.

---
**Algorithm 2 DO-EM** for **CQ-LVM**

---
1: **Input:** Target density operator $\eta_{\mathrm{V}}$ and $\theta^{(0)}$
2: **while** not converged **do**
3: $\quad \mathcal{Q}_i(\theta; \theta^{(k)}) = \mathrm{Tr}\left(\rho_{\mathrm{L}}^{(i)}(\theta^{(k)})e^{\mathrm{H}^{(i)}(\theta)}\right) - \log Z(\theta)$
4: $\quad \theta^{(t+1)} = \mathrm{argmax}_{\theta} \frac{1}{N}\sum_{i=1}^{N} \mathcal{Q}_i(\theta; \theta^{(k)})$

---

### 5.1 Quantum Boltzmann Machine

267 In this section, we discuss the QBM and define variants which are amenable to implementation on
268 high-dimensional classical data. We first describe QBMs that are **CQ-LVMs**.

269 **Corollary 5.5.** *A* $\mathrm{QBM_{m,n}}$ *is a **CQ-LVM** if and only if quantum terms on the visible units are zero.*

270 *Proof sketch:* The statement is true because of the structure of Pauli matrices which have entries
271 outside the direct sum structure if and only if $i \leq m$. A detailed proof can be found in Appendix C.

272 The class of semi-quantum models studied in Demidik et al. [33] are **CQ-LVMs**. Training such a
273 QBM is intractable for real-world data since the free energy term, $-\log Z(\theta)$ is intractable even for
274 classical Boltzmann machines. To achieve tractable training of QBMs, we introduce the **Q**uantum
275 **I**nterleaved **D**eep **B**oltzmann **M**achine (QiDBM) that can be trained using Contrastive Divergence with
276 a quantum Gibbs sampling step derived here.

277 A **Q**uantum **I**nterleaved **D**eep **B**oltzmann **M**achine (QiDBM) is a DBM with quantum bias terms on
278 **non-contiguous hidden layers**. We describe the Hamiltonian of a three-layered $\mathrm{QiDBM_{\ell,m,n}}$ with $\ell$
279 visible units and $m$ and $n$ hidden units respectively in the two hidden layers. For ease of presentation,
280 the quantum bias terms are present in the middle layer.

$$\mathrm{H} = -\sum_{i=1}^{\ell+m+n} b_i\sigma_i^z - \sum_{i=1}^{\ell}\sum_{j=1}^{m} w_{ij}^{(1)}\sigma_i^z\sigma_{\ell+j}^z - \sum_{i=1}^{m}\sum_{j=1}^{n} w_{ij}^{(2)}\sigma_{\ell+i}^z\sigma_{\ell+m+j}^z - \sum_{i=1}^{m}\Gamma_i\sigma_{\ell+i}^x \qquad \text{(QiDBM)}$$

281 The quantum interleaving in a QiDBM is necessary to make the Gibbs sampling step tractable. We
282 illustrate the case of the middle layer of $\mathrm{QiDBM_{\ell,m,n}}$. If the non-quantum visible and hidden layers
283 are fixed to $\mathbf{v}$ and $\mathbf{h}^{(2)}$, the hidden units of the quantum layer are conditionally independent. The
284 Hamiltonian of the $i^{\mathrm{th}}$ unit of the quantum layer $\mathrm{L}^{(1)}$ is given by $\mathrm{H}^{\mathrm{L}^{(1)}}(i|\mathbf{v}, \mathbf{h}^{(2)}, \theta) = -b_i^{\mathrm{eff}}\sigma^z - \Gamma_i\sigma^x$.
285 This allows for the tractable sampling from the quantum layer using the expected values

$$\langle\sigma_i^z\rangle_{\mathbf{v}, \mathbf{h}^{(2)}} = \frac{b_i^{\mathrm{eff}}}{D_i}\tanh D_i \text{ and } \langle\sigma_i^x\rangle_{\mathbf{v}, \mathbf{h}^{(2)}} = \frac{\Gamma_i}{D_i}\tanh D_i$$

286 where $D_i = \sqrt{(b_i^{\mathrm{eff}})^2 + \Gamma_i^2}$ and $b_i^{\mathrm{eff}} = b_i + \sum_{j=1}^{\ell} w_{ij}^{(1)}\mathbf{v}_j + \sum_j w_{ij}^{(2)}\mathbf{h}_j^{(2)}$. The Gibbs step for the
287 non-quantum layers is done as per the classical CD algorithm using the quantum sample from the $Z$
288 Pauli operator. This closed-form expression for Gibbs sampling without matrices allows CD to run
289 on a QiDBM with the same memory footprint as a DBM. See Appendix C for more details.

## 6 Empirical evaluation

291 In this work, we propose a quantum model **CQ-LVM**, and a general EM framework, **DO-EM**, to
292 learn them. In this section, we empirically evaluate our methods through experiments to answer
293 the following questions. Details of the compute used to run all our experiments and baselines are
294 provided in Appendix D and E.

295 (Q1) **Effectiveness of DO-EM.** Is Algorithm 2, a feasible algorithm for **CQ-LVM**s compared to
296 $\quad$ state of the art algorithms for QBMs ?

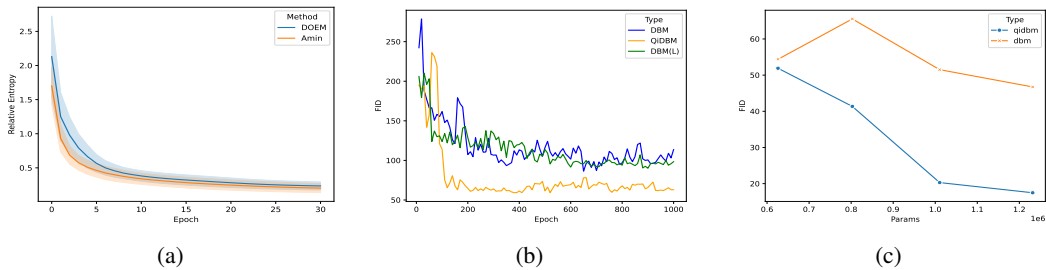

Figure 1: (a) Relative entropy during training with exact computation of a QBM on a mixture of Bernoulli distribution. Showing that DO-EM does lead to decrease in relative entropy. (b) DBM with 6272 hidden units. QiDBM with 6273 hidden units. DBM(L) with 6273 hidden units. (c) FID scores on Binarized MNIST as a function of model parameters of QiDBM and DBM.

(Q2) **DO-EM on Real World Data.** Does Algorithm 2 scale with the to real world data?

(Q3) **Performance of DO-EM.** Does Algorithm 2 provide reasonable improvement in performance over classical LVMs?

To answer (Q1), we conduct experiments running exact computation to show that the proposed algorithm is feasible and is practical to implement.

**Baselines** We compare our method with our implementation of Amin et al. [12] which explores an alternate algorithm for training QBMs.

**Dataset and Metrics** We use a mixture of Bernoulli dataset introduced in Amin et al. [12] described in Appendix D. We measure the efficacy of our proposed method by measuring the average relative entropy during training.

**Results of experiment** In Figure 1a, we first observe that the relative entropy of our proposed algorithm does decrease during training, validating our theoretical results and showing, to the best our knowledge, the first instance of an expectation maximization algorithm with quantum bias. We also observe that the performance is competitive with Amin et al. [12]. We also note that **CQ-LVM** training with DO-EM is faster than Amin et al. [12] and consumes lesser memory. We provide more experiments using exact computation in Appendix D.

To answer (Q2) and (Q3), we conduct experiments on DBMs of varying sizes with and without the quantum bias term described in Section 5. We present qualitative results of our experiments in Appendix D.

**Baselines.** We compare our proposed method with Taniguchi et al. [22], the state of the art for training DBMs. We are unable to reproduce the results in their work and we report the results obtained from their official implementation[1] using the hyper parameters described in their work.

**Datasets and Metrics** Following prior work [22], we perform our experiments on MNIST and Binarized MNIST dataset [44] which contains 60,000 training images and 10,000 testing images of size 28x28. We measure the FID [45] between 10,000 generated images and the MNIST test set to assess the quality of generation. The Fréchet Inception Distance (FID) is a quantitative metric used to evaluate the quality of images generated by generative models by comparing the statistical distribution of their feature representations to those of real images.

**Experiment: Performance of DO-EM** To show the superior performance of the proposed method, we compare the FID of our proposed algorithm on Binarized MNIST. We train a QiDBM and DBM with 498, 588, 686, and 784 hidden units with a learning rate of 0.001 for 1000 epochs with 2 hidden layers with SGD optimizer with a batch size of 600.

**Results of Experiments** In Figure 1c, we observe that the proposed algorithm outperforms the DBM in all cases, achieving a minimum FID of 14.77 to the DBM's 42.61. This experiment shows that simply adding quantum bias terms to a DBM can *improve the quality* of generations by around 65%.

---

[1] https://github.com/iShohei220/unbiased_dbm

**Experiment: DO-EM on High Dimensional Data** We run CD on 2 DBMs without quantum bias terms according to Taniguchi et al. [22] and CD with quantum bias for a QiDBM on MNIST. Each image corresponds to 6272 visible binary units. The QiDBM has 78.70M parameters with 2 hidden layers with quantum bias added to the second layer with a hidden size of 6272. Both DBMs have 2 hidden layers and have 78.69M and 78.71M parameters and hidden sizes of 6272 and 6273 respectively. We use a learning rate of 0.001 for all experiments and train with a batch size of 600 with SGD optimizer for 1000 epochs. The purpose of this experiment is to show that it is feasible to train large models with quantum bias terms.

**Results of Experiments** In Figure 1b, we observe that the proposed method outperforms both classical models of similar size with a 45% reduction in FID. We observe that the FID of the model converges to this value in around 400 epochs whereas both DBM models still exhibit instability after 500 epochs. The QiDBM achieves an FID of 62.77 whereas the classical DBMs achieve an FID of 111.73 and 99.17 for the smaller and larger model respectively. This experiment indicates that scaling QiDBMs is feasible and provides a significant improvement in performance. In Appendix D, we show the qualitative differences between generated samples of the DBM and QiDBM. We observe that the generated samples from the QiDBM appear to be better than that of the DBM after only 250 epochs.

**Discussion** We design **CQ-LVM**s and implement Algorithm 1 to learn different target distributions. We first show that Algorithm 1 is effective in learning **CQ-LVM**s and is competitive with the state of the art in terms of reduction of relative entropy at lower running times for 10 qubits and can be extended to even 20 qubits where others cannot. Next, we see that the addition of quantum bias terms to a DBM when trained using Algorithm 2 shows superior generation quality compared to classical DBMs with a 60% reduction of FID on Binarized MNIST. Next, we show that **QiDBMs** can learn high dimensional datasets like MNIST using Algorithm 2 by scaling models upto 6272 hidden units. We observe that QiDBMs also achieve better performance, with 40% lower FID compared to DBMs of similar sizes. We also observe that QiDBMs converge in about half the amount of time compared to DBMs.

# 7    Discussion

The paper makes important progress by proposing **DO-EM**, an EM Algorithm for Latent Variable models defined by Density Operators, which provably achieves likelihood ascent. We propose **CQ-LVM**, a large collection of density operator based models, where **DO-EM** applies. We show that QiDBM, an instance of **CQ-LVM**, can easily scale to MNIST dataset which requires working with 6200+ units and outperform DBMs, thus showing that Density Operator models may yield better performance. The specification of **DO-EM** is amenable to implementation on quantum devices.

**DO-EM on quantum devices** The E Step of the DO-EM algorithm can be implemented on a quantum computer using the method developed by Gilyén et al. [46], where the quantum channel is performing the partial trace operation. The goal is to prepare the Petz recovery map for the partial trace channel $\eta^{(t)} = \mathcal{R}_\rho(\eta_V)$ using PRM. The requirements for this are (1) Quantum access to the input state $\eta_V$ (2) efficient state preparation of the model's density matrix $\rho(\theta)$ [47, 48] and (3) Block-encodings for the model's density matrix and its marginal $\rho_V(\theta) = \text{Tr}_L\rho(\theta)$ [49]. Given these input assumptions, the quantum algorithm implementing PRM consists of three steps [46]: (1) applying $\rho_V^{-1/2}$ on the state $\eta_V$, (2) applying the adjoint channel which is straight-forward for the partial trace channel and can be operationally achieved by preparing subsystem L in the maximally mixed state, and (3) applying $\rho^{1/2}$ on the combined system. Both $\rho_V^{-1/2}$ and $\rho^{1/2}$ are implemented using *Quantum Singular Value Transformation (QSVT)* techniques, leveraging block-encodings of the relevant states [49].

The M Step proceeds via gradient descent by the computation of the gradient given by $\left( \text{Tr}[H_r \eta(\theta^{(t)})] - \text{Tr}[H_r \rho(\theta)] \right)$ for the different terms in the Hamiltonian $H = \sum_r \theta_r H_r$ [14, 32]. The M Step stops when the gradients are small and an updated parameter $\theta^{(t+1)}$ is obtained. This two-step iterative DO-EM procedure continues until convergence. While the gradients can be estimated on existing near-term quantum devices, the E step requires careful design.

**Limitations** We discuss the limitations of this work in Appendix F.

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
