# OpenReview forum: "DO-EM: Density Operator Expectation Maximization"
_NeurIPS.cc/2025/Conference — Submitted to NeurIPS 2025_

### Official Review · Reviewer_W6WS · 2025-06-29

**Clarity:** 3
**Significance:** 2
**Originality:** 4
**Rating:** 4
**Confidence:** 4

**Summary:**

This paper studies density operator models, which existing training algorithms can't scale to real data. The paper considers a new training algorithm, called Density Operator Expectation Maximization (DO-EM), which overcomes some existing issues by formulating the expectation set as QIP in a novel way. The author also proposes a new generative modeling algorithm, Quantum Interleaved Deep Boltzmann Machines (QiDBMs), and empirically evaluate it MNIST data.

**Questions:**

Most of my concerns are raised in the weakness section. However, I do have some more questions that could use some clarification, which I have listed below.

1. Relationship between terms. I am a bit confused about the relationship between DO-LVM, CQ-LVM, and QiDBM, and it would be great if you could elaborate more on that.

2. Similarly, what's the connection between the proposed DO-EM and the contrastive divergence algorithm used in practice? Are they the same thing when applied to QiDBM?

3. Comparison with other quantum generative modeling algorithms. I understand that benchmarking on MNIST might be a standard practice in the community at this current stage of progress. However, to show that the proposed algorithm has an advantage, it should at least be compared to other quantum ML algorithms on the same task. If this can't be done, what's the major difficulty that prevents you from doing so?

I am happy to increase my recommendation for the work if the experiments section is enhanced to demonstrate the superiority of the proposed method, and the related work section is added for better readability.

**Ethical Concerns:**

["NO or VERY MINOR ethics concerns only"]

**Final Justification:**

The paper provides a thorough, rigorous theoretical study of the learning for Quantum Density Operator. Although the studied quantum algorithm is novel, the empirical validation is relatively limited with non-satisfactory performance. Therefore, I recommend borderline acceptance.

**Limitations:**

yes, limitations are addressed.

**Quality:**

3

**Strengths And Weaknesses:**

**Strength:**
1. The presented algorithm is novel.
2. The proof and derivation details appear to be technically sound (although I did not thoroughly verify them).
3. The experiment includes real datasets to demonstrate practical relevance.


**Weakness**
1. The paper misses a detailed literature review related to quantum generative modeling algorithms, for example [1,2,3,4], and as well as those classical algorithms for generating quantum data, for example [5,6]. I believe the list is not exhaustive, as this is a fast-growing field. Presenting a clear overview of the literature and explaining in detail the advantages of the proposed algorithm compared to existing ones is essential for readers to understand the contribution and significance of this paper.

2.  Missing important benchmarks in empirical validation. The experiment section only includes comparison with DBM, while potentially missing other quantum generative modeling algorithms. If these algorithms are not suitable for the task, a paragraph should at least be included to explain the reason.

3. Minor practical relevance. The paper only experiments on MNIST, while the best result only achieves an FID of 20. From a practitioner's perspective, this suggests a highly inferior performance compared to those classical generative modeling algorithms. A clear explanation of why the proposed quantum algorithms should be favored over these classical algorithms should be added to the introduction to better motivate the work. Moreover, if the experiments can be conducted on CIFAR-10 or an extremely downscaled version of ImageNet (e.g., ImageNet-32 or even smaller), the quality of the paper will be greatly enhanced.


**References**

[1] Dallaire-Demers, Pierre-Luc, and Nathan Killoran. "Quantum generative adversarial networks." Physical Review A 98.1 (2018): 012324.

[2] Zhang, Bingzhi, et al. "Generative quantum machine learning via denoising diffusion probabilistic models." Physical Review Letters 132.10 (2024): 100602.

[3] Riofrio, Carlos A., et al. "A characterization of quantum generative models." ACM Transactions on Quantum Computing 5.2 (2024): 1-34.

[4] Kwun, Gino, Bingzhi Zhang, and Quntao Zhuang. "Mixed-state quantum denoising diffusion probabilistic model." Physical Review A 111.3 (2025): 032610.

[5] Zhu, Yuchen, et al. "Quantum state generation with structure-preserving diffusion model." arXiv preprint arXiv:2404.06336 (2024).

[6] Hibat-Allah, Mohamed, et al. "A framework for demonstrating practical quantum advantage: comparing quantum against classical generative models." Communications Physics 7.1 (2024): 68.

---

> ### Author Rebuttal · Authors · 2025-07-31
>
> We thank the reviewer for the review. We first answer the questions raised by the reviewer:
>
> 1. **Relationship between terms. I am a bit confused about the relationship between DO-LVM, CQ-LVM, and QiDBM, and it would be great if you could elaborate more on that.**
>
> Density Operator Latent Variable Models (DO-LVM) are quantum analogues of classical Latent Variable Models. These models can be used to learn both quantum and classical data. Our results in Section 4 establish that the QELBO is a minorant of the log-likelihood function if the model satisfies Condition S. In Section 5, we focus on using DO-LVMs to train on classical data. We show that DO-LVMs that satisfy Condition S for classical data have a Classical visible space and a Quantum hidden space. We call such models CQ-LVMs. To empirically test these models and our algorithm on classical data, we introduce the Quantum Interleaved Deep Boltzmann Machine (QiDBM), an example of a CQ-LVM.
>
> 2. **Similarly, what's the connection between the proposed DO-EM and the contrastive divergence algorithm used in practice? Are they the same thing when applied to QiDBM?**
>
> It is well known that computing the expectation terms required for training Restricted Boltzmann Machines (RBMs) exactly is intractable, due to the exponential sum over the model distribution. The breakthrough  Contrastive Divergence (CD) algorithm was able to approximate these expectations using a few steps of Gibbs sampling initialized at the data, enabling RBMs to be trained efficiently and unlocking their widespread use in unsupervised learning and pretraining deep networks. Quantum Deep Boltzmann Machines face an analogous problem, preventing gradient ascent techniques in the M step of the DO-EM algorithm. The Contrastive Divergence update used to train QiDBMs approximates the necessary expectation values for the M step and is the Density Operator analogue of the classical Contrastive Divergence technique.
>
> 3. **Comparison with other quantum generative modeling algorithms. I understand that benchmarking on MNIST might be a standard practice in the community at this current stage of progress. However, to show that the proposed algorithm has an advantage, it should at least be compared to other quantum ML algorithms on the same task. If this can't be done, what's the major difficulty that prevents you from doing so?**
>
> The quantum generative algorithms pointed out by the reviewer simulate generative models on a Parametrized Quantum Circuit (PQC). These efforts are orthogonal to the efforts in our paper where we develop algorithms to train density operator models agnostic to their implementation on a specific kind of quantum or classical device. Moreover, these existing models cannot train or generate more than 12-dimensional binary bitstrings (please see next question). This clearly shows that SoTA is far from handling real world data; even MNIST size data (which is a 6000+ dimensional binary bitstring) is significantly out of the ball-park.
>
> The primary advantage of our algorithm as compared to existing quantum algorithms is scalability. Our algorithm is able to train 6272-dimensional datasets (a 500x increase in dimensionality) compared to existing algorithms. Hence, we establish the first benchmarks of a quantum algorithm on generating real-world data. We request the reviewer to go through point 4 as well.
>
> We now address the weaknesses.
>
> 4. **The paper misses a detailed literature review related to quantum generative modeling algorithms, for example [1,2,3,4], and as well as those classical algorithms for generating quantum data, for example [5,6]. I believe the list is not exhaustive, as this is a fast-growing field. Presenting a clear overview of the literature and explaining in detail the advantages of the proposed algorithm compared to existing ones is essential for readers to understand the contribution and significance of this paper.**
>
> Quantum Machine Learning has been broadly used to mean quantumness either in the model or in the data or both. In the references pointed out, the models are defined by a parametrized quantum circuit (PQC), which are known to suffer from training (barren plateaus) and scalability issues (<= 12 qubits; see lines 46-48). The numerics in [1,2,3,4] are respectively demonstrated over 5, 4, 12, and 12 qubits. This work advances a complementary approach using Density Operators. A general framework to learn Density Operators is proposed here, significantly advancing the spirit of Warmuth and Kuzmin Ref. [34]. This work successfully grapples with the quantum challenge of non-commutativity and paves the way for algorithmic advances by deriving the Quantum Evidence Lower Bound (QELBO). We show an unprecedented scale (>6000 qubits) by exploiting the structure in the Hamiltonian.
>
> The benchmark comparison with an earlier quantum generative model by Amin et al. Ref. [12] is included in Fig 1(a), over a dataset defined over 10 qubits.
>
>
> 5. **Missing important benchmarks in empirical validation. The experiment section only includes comparison with DBM, while potentially missing other quantum generative modeling algorithms. If these algorithms are not suitable for the task, a paragraph should at least be included to explain the reason.**
>
> Please refer to points 3 and 4. We thank the reviewer for the suggestion and shall update our manuscript to clarify the positioning of our work concerning these papers, as pointed out in the previous responses.
>
> 6. **Minor practical relevance. The paper only experiments on MNIST, while the best result only achieves an FID of 20. From a practitioner's perspective, this suggests a highly inferior performance compared to those classical generative modeling algorithms. A clear explanation of why the proposed quantum algorithms should be favored over these classical algorithms should be added to the introduction to better motivate the work. Moreover, if the experiments can be conducted on CIFAR-10 or an extremely downscaled version of ImageNet (e.g., ImageNet-32 or even smaller), the quality of the paper will be greatly enhanced.**
>
> We do not directly compare ourselves with SOTA generative models which use deep neural networks. This work focuses on developing the theoretical and algorithmic framework for doing Machine Learning with Density Operators - a nascent field gaining importance due to the possibilities of quantum hardware. There are no algorithms that can train Density Operator models on real-world data. We bridge this gap and compare ourselves to the SOTA deep Boltzmann machine, the closest possible comparison, and show demonstrable improvements. Our work showcases that Density Operators are a promising area of future machine learning research and believe that Density Operator analogues of DNN based generative models will provide a significant advantage.
>
> On the reviewer’s request we provide new experiments in Question 2 of the rebuttal to reviewer BprH. We request the reviewer to review at the same.
>
> **I am happy to increase my recommendation for the work if the experiments section is enhanced to demonstrate the superiority of the proposed method, and the related work section is added for better readability.**
>
> We thank the reviewer for their comprehensive review. Your valuable review has helped us strengthen our work, specifically in providing context for related work in this field and enhancing the practical applicability of our work. We hope we have answered your questions adequately and are happy to provide further clarification.

---

> > ### Comment · Reviewer_W6WS · 2025-08-02
> >
> > Thank you for the response! I have better recognized the contribution of the paper, despite its relatively non-satisfactory empirical validation. I have increased my recommendation for the work, given the promised revision in the updated manuscript as well as the additional numerical results.

---

> ### Author Response · Authors · 2025-08-03
> **Experimental results on CelebA-32 with Quantum Gaussian RBMs**
>
> We thank the reviewer for the response and for increasing your evaluation of the paper. **We showcase additional experimental results on a downscaled 32x32 CELEBA dataset on a Quantum Gaussian RBM achieving 40-50% relative improvement compared to a classical Gaussian RBM after 250 epochs.**
>
> In our submission, we developed the Contrastive Divergence algorithm for the Quantum analog of the Deep Boltzmann Machine to learn the MNIST dataset. To train on more complicated datasets, we have **extended the DO-EM algorithm and Contrastive Divergence to Quantum analogues of Gaussian RBMs (GRBM)** [1-2]. GRBMs are significantly more complicated models than RBMs and DBMs with continuous visible variables and binary hidden variables. We train Quantum and Classical GRBMs on a downscaled CelebA dataset resized to 32x32 pixels.
>
> We observe that the quantum variant achieves 40-50% lower FID than a GRBM of the same hidden dimension for the same amount of training time. **After 250 epochs, the GRBM achieves an FID of 245.77, and the QGRBM achieves an FID of 151.33, a ~40% decrease. After 600 epochs, the GRBM achieves an FID of 134.71 compared to the QGRBM, which achieves an FID of 59.83, a ~50% decrease.** We will make all code available as allowed per conference guidelines.
>
> As noted in question 6 in our rebuttal, the purpose of this work is not to compete with SoTA deep neural architectures but to **establish the theoretical and algorithmic foundation of Density Operator Models**, an area gaining prominence due to the emergence of quantum technology. **Our new experiments show that our proposed methods can significantly outperform their classical counterparts across models and datasets.**
>
> [1] Welling, M., Rosen-Zvi, M., & Hinton, G. E. (2004). Exponential family harmoniums with an application to information retrieval. Advances in neural information processing systems, 17.
> [2] Liao, R., Kornblith, S., Ren, M., Fleet, D. J., & Hinton, G. (2022). Gaussian-bernoulli rbms without tears. arXiv preprint arXiv:2210.10318.

---

### Official Review · Reviewer_FuJL · 2025-07-01

**Clarity:** 3
**Significance:** 3
**Originality:** 3
**Rating:** 4
**Confidence:** 4

**Summary:**

The paper proposes a quantum inspired EM algorithm for density operators that replaces the classical E step by a quantum relative entropy projection step.

The algorithm is placed in a DO-LVM (density operator latent variable model) framework that generalizes classical latent variable models to density operators using partial traces to separate the observable and latent variables. This is a natural definition and leads also to quantum analogs of RBMs. However, these models are harder to train classically due to the novel quantum relative entropy projection and their advantage is not clear priori. Experiments to determine the advantage of the classical models are performed and improvements in quality are observed with respect to a specific metric.

A quantum realization of the EM algorithm is proposed in principle, but the details are not fully worked out, so it unclear if this is a viable algorithm on a quantum computer.

**Questions:**

- Line 172: Is it Theorem 2.2 being proved or Lemma 4.1?

- It would be helpful to have more explanations on the new ideas introduced in this work like ELBO, minorant maximization and Petz recovery map in the main body of the paper. Most of the new ideas and proofs are found in the appendix due to space constraints, but I would suggest that  a sketch of the novel ideas in the E step and brief proof sketches should be added if the paper is accepted.

**Ethical Concerns:**

["NO or VERY MINOR ethics concerns only"]

**Final Justification:**

The paper woud benefit from a substantial revision addressing the quastions raised during the review process. I am therefore maintaining my score of 4.

**Limitations:**

Discussed adequately.

**Paper Formatting Concerns:**

None.

**Quality:**

3

**Strengths And Weaknesses:**

- The idea for implementing the E step using quantum information projections and relating to the Petz recovery map is novel and goes beyond the existing QML literature on clustering and latent variable models.

There has been previous work on quantum expectation maximization (arXiv:1908.06657) but progress has been limited as it quantizes the classical subroutines used in the classical EM algorithm for Gaussian mixtures. The present work however does not give an end-to-end quantum algorithm but introduces a novelty in the E-step.

- The DO-LVM framework is a natural quantum framework generalizing classical LVMs and investigating advantages it may offer is worthy of further investigation. However, it is harder to train than the classical LVMs and the possible advantages are in qualtity and scalability. The experiments report a significantly improved Frechet Inception distance (FID) for the classical models (CQ-LVM) over Boltzmann machines, this is a metric that shows that the images being generated are more 'realistic' and is thus a measure of improved quality. It is not clear however, if the FID is competitive with state of the art image generators.

- The end-to-end quantum EM algorithm needs to be specified in greater detail, specially the input model and the implementation of the quantum E step with SVT. A resource analysis should be carried out to determine how well this scales with qubits and circuit size to see if this is indeed a viable method for early quantum computers.

The paper introduces novel ideas that quantize one of the steps in the EM algorithm in a non trivial way and is therefore worthy of serious consideration. However, some parts need to be worked out further or explained better so I am going with a weak accept.

---

> ### Author Rebuttal · Authors · 2025-07-31
>
> We thank the reviewer for the review and discuss the points raised below.
>
> 1. **Line 172: Is it Theorem 2.2 being proved or Lemma 4.1?**
>
> We use Theorem 2.2, a well known result in quantum information theory, to prove Lemma 4.1. The proof is in the Appendix.
>
> 2. **It would be helpful to have more explanations on the new ideas introduced in this work like ELBO, minorant maximization and Petz recovery map in the main body of the paper. Most of the new ideas and proofs are found in the appendix due to space constraints, but I would suggest that a sketch of the novel ideas in the E step and brief proof sketches should be added if the paper is accepted.**
>
> We thank the reviewer for noting the novelty of our results and shall suitably add commentary on the proofs if the paper is accepted.
>
> Currently, quantum generative models are restricted to the QBM. Existing work on the QBM is unable to train on high-dimensional, real-life datasets such as MNIST and barely scale beyond 12 qubits. This is because of the fundamental difficulty in learning DO-based latent variable models (LVM). We derive a generic algorithm for DO-LVMs in the spirit of the Evidence Lower Bound and the highly influence EM algorithm. We bring together results from information geometry, operator theory, quantum information theory, and the classical tenets of ML to derive an EM algorithm for density operators. We wish to point out some advantages of our work which were not possible previously:
>
> - Operator-theoretic versions of Jensen’s inequality are ill-suited for deriving a Quantum Evidence Lower Bound (QELBO). To address this, we instead leverage the Monotonicity of Relative Entropy—a fundamental result in quantum information theory—applying it, for the first time, to the training of density operator models.
> - The E Step in the classical EM algorithm is achieved using conditional probability. There is no well-defined notion of conditional operators given a joint density matrix. We use an information geometric interpretation of classical EM provided in to develop DO-EM. To this end, we develop the formalism around Quantum Information Projection (QIP). To the best of our knowledge, this is new and of independent interest.
> - We show that the QIP under certain sufficiency conditions (Condition S) is the well-known Petz Recovery Map (PRM) and obtain a closed-form expression for the E-step. The PRM is a quantum channel in our setting, paving the way for implementing DO-EM on a quantum device. This result reveals connections between DO-LVMs and quantum information theory – paving the way for further exploration of these models.
> - Using the results above, we guarantee an increase in the likelihood after each iteration of DO-EM, a key-property of the classical EM algorithm.
> - Moreover, we demonstrate that both the E step and the M step are significantly easier to implement than direct gradient maximization of the log-likelihood. This is analogous to the EM algorithm changing a log-of-sums maximization problem to a sum-of-logs maximization problem.
>
> Our work is a generic framework to train any DO-LVM. Similar to the classical ELBO, we provide an approach to train any DO-based generative model. We believe that this will give impetus to richer DO-based models.
>
> We now address the weakness raised by the reviewer.
>
> 3. **The DO-LVM framework is a natural quantum framework generalizing classical LVMs and investigating advantages it may offer is worthy of further investigation. However, it is harder to train than the classical LVMs and the possible advantages are in quality and scalability.**
>
> We request the reviewer to clarify their statement. As the reviewer rightly points out, we experimentally show that DO-LVMs generate better images than classical LVMs. The Gibbs Sampling step derived in line 285 allows the training of QiDBMs on the same hardware and resources as a classical DBM. While the formalism is new and challenging, our theoretical results make the training of DO-LVMs as easy as their classical counterparts.
>
> 4. **The experiments report a significantly improved Frechet Inception distance (FID) for the classical models (CQ-LVM) over Boltzmann machines, this is a metric that shows that the images being generated are more 'realistic' and is thus a measure of improved quality. It is not clear however, if the FID is competitive with state of the art image generators.**
>
> We acknowledge that the FID of our model does not yet match that of state-of-the-art image generators such as GANs or diffusion models. However, our goal is not to compete directly with these architectures on generation quality. Rather, we focus on developing a scalable algorithmic framework for training Density Operator Latent Variable Models (DO-LVMs), which are increasingly relevant given the rise of quantum hardware. Existing DO-based models fail to scale beyond ~12 qubits due to the lack of general training algorithms. In contrast, we introduce a method that scales to 6272 qubits—a 500× increase in Hilbert space dimension—allowing DO models (e.g., Quantum DBMs) to operate at the same scale as classical Boltzmann machines. Moreover, our results show that Quantum DBMs consistently outperform classical DBMs on the same hardware and resources. This establishes DO models as a viable and promising class for future quantum-accelerated machine learning, independent of current FID benchmarks. We view our work as laying the groundwork—analogous to early Boltzmann machine papers that preceded modern deep generative models.
>
> 5. **The end-to-end quantum EM algorithm needs to be specified in greater detail, specially the input model and the implementation of the quantum E step with SVT. A resource analysis should be carried out to determine how well this scales with qubits and circuit size to see if this is indeed a viable method for early quantum computers.**
>
> A detailed end-to-end implementation of the EM algorithm on a quantum device is beyond the scope of this work and remains a substantial direction for future research. Such an analysis would involve several considerations including quantum data access and noise assumptions being made. Quantum data access is a subtle and important point requiring careful consideration [1]. Presently, the QSVT routine is a subject of intense research efforts and viable end-to-end quantum algorithms may soon be a possibility [2]. In this work we develop the mathematical and algorithmic framework to train Density Operator Models. We can only provide a sketch of an end-to-end quantum algorithm at this stage, to highlight that the DO-EM algorithm is amenable to implementation on quantum devices. However, we leave its development as future work.
>
> 6. **The paper introduces novel ideas that quantize one of the steps in the EM algorithm in a non trivial way and is therefore worthy of serious consideration. However, some parts need to be worked out further or explained better so I am going with a weak accept.**
>
> We thank the reviewer for their thoughtful feedback and for engaging seriously with our submission. We hope this rebuttal has addressed your concerns, and we welcome any further suggestions that would help strengthen the paper or improve your evaluation.
>
> [1] Scott Aaronson, “Read the fine print”, Nature Physics 11, 291-293 (2015)
> [2] Chakraborty et al., “Quantum singular value transformation without block encodings: Near-optimal complexity with minimal ancilla”, Preprint arxiv:2504.02385 (2025)

---

> > ### Comment · Reviewer_FuJL · 2025-08-02
> >
> > Thank you for your comments and clarifications, particularly on question 2. As the other reviewers also note, some revisions would help enhance the clarity of the manuscript. I am maintaining my score of 4.

---

### Official Review · Reviewer_fThS · 2025-07-02

**Clarity:** 2
**Significance:** 3
**Originality:** 3
**Rating:** 3
**Confidence:** 2

**Summary:**

The authors propose Density Operator Expectation Maximization, an expectation maximization algorithm for latent variable models. Empirically, they demonstrate their idea on the MINIST dataset, showing clear performance improvements. The results imply the better scalability of the method on Boltzmann quantum operators.

**Questions:**

I am not sure how the proposed algorithm is closely aligned with latent variable models from the algorithm block.

**Ethical Concerns:**

["NO or VERY MINOR ethics concerns only"]

**Final Justification:**

I appreciate the authors’ efforts in preparing the rebuttal. However, the results presented are still preliminary. Therefore, I have decided to maintain my score.

**Quality:**

2

**Strengths And Weaknesses:**

Strengths: The paper has provided theoretical analysis of the algorithm. The empirical experiments on MNIST show large performance gain.

Weakness:
- My main concern of the paper is its accessibility to the audience. With the overuse of shorthand notations, the paper sometimes is difficult to follow.

- Line 87: I am not sure why "this can only discover local minima". This claim seems to be disconnected from the context above and not discussed clearly.

- Algorithm 1: It would be better if the authors can provide comments for each step, and the notations are very confusing. I am not sure what does it mean to have the target operator $\eta_V$ as the input.

- Contribution 3: why name this as CQ?

- Why do we need the contrastive divergence for the training?

- Figure 1: Why we cannot have the consistent name of the proposed method?

- Experiments on MNIST: Why each image correspond to 6272 binary units?

- Discussion: Could we use consistent notations? For example, you can keep the capitalization of the latent variable models or not. But they should be consistent.

---

> ### Author Rebuttal · Authors · 2025-07-31
>
> We thank the reviewer for their review.
>
> The reviewer’s summary describes Density Operator Expectation Maximization as an expectation maximization algorithm for latent variable models. We would like to clarify that the proposed algorithm is not for Probabilistic Latent Variable Models but for LVMs defined through Density Operators. Density Operators are quantum generalizations of probability distributions and is gaining importance in machine learning due to the emergence of quantum computing. Currently, there are no generic algorithms to train Density Operator Latent Variable Models. In this light, we request the reviewer to go through the following points.
>
> 1. **My main concern of the paper is its accessibility to the audience. With the overuse of shorthand notations, the paper sometimes is difficult to follow.**
>
> With the advent of quantum computing devices, the problems addressed in this paper are more important than ever for the AI community. We are happy to edit the notation for better readability if given concrete suggestions.
>
> 2. **Line 87: I am not sure why "this can only discover local minima". This claim seems to be disconnected from the context above and not discussed clearly.**
>
> This statement is relevant in the context of any latent variable model. To clarify, gradient-based algorithms when applied to minimization of negative log-likelihood of a latent variable model, usually leads to a local optimum because of the non-convexity of the objective (negative log-likelihood).
>
> 3. **Algorithm 1: It would be better if the authors can provide comments for each step, and the notations are very confusing. I am not sure what does it mean to have the target operator as the input.**
>
> Commentary for Algorithm 1 is given in **lines 213-222** after the associated theoretical results are presented. The Algorithm 1 has only 2 steps (E Step and M Step) which are iterated until convergence.
>
> As mentioned in Definition 3.1, Density Operators Latent Variable Models try to learn a target density operator, which is the quantum mechanical generalization of a probability distribution. Just as a Latent Variable model tries to learn the joint probability distribution over observed and latent variables that best explains the given input data (the target), a DO-LVM learns a joint density operator best approximating the input data (either classical or quantum data encoded as a target density operator). A target density operator for classical data is provided in **line 234**.
>
> 4. **Contribution 3: why name this as CQ?**
>
> DO-LVMs may train on both classical or quantum data. In Section 5, we specialize DO-LVMs to classical data. As mentioned in line 241, Density Operator LVMs that satisfy Condition S on classical data have visible units that behave Classically (the quantum bias is set to zero) and hidden units that behave in a Quantum manner (non-zero quantum bias). The name CQ conveys this. In the equation labeled (QiDBM) after **line 280**, the quantum bias term is shown in blue.
>
> 5. **Why do we need the contrastive divergence for the training?**
>
> It is well known that computing the expectation terms required for training Restricted Boltzmann Machines (RBMs) exactly is intractable, due to the exponential sum over the model distribution (**lines 94-99**). Hinton's breakthrough Contrastive Divergence (CD) algorithm [18,19 in the main paper] was able to approximate these expectations using a few steps of Gibbs sampling initialized at the data, enabling RBMs to be trained efficiently and unlocking their widespread use in unsupervised learning and pretraining deep networks. Quantum Deep Boltzmann Machines face an analogous problem, preventing gradient ascent techniques in the M step of the DO-EM algorithm (see **lines 262-265 and 272-274**). The Contrastive Divergence update used to train QiDBMs approximates the necessary expectation values for the M step and is the Density Operator analogue of the classical Contrastive Divergence technique.
>
> 6. **Experiments on MNIST: Why each image correspond to 6272 binary units?**
>
> As mentioned in line 316, we follow the baselines established in Ref. [22] (Taniguchi et al., ICML 2023), the SoTA paper on training DBMs. The method in Ref. [22] takes an 8 bit image (0-255 values) and binarizes each pixel (0 or 1 values). The binarization takes an 8 bit MNIST image to 8x28x28 = 6272 binary units.
>
> 7. **I am not sure how the proposed algorithm is closely aligned with latent variable models from the algorithm block.**
>
> This is explained in **lines 196-201**. In the E step, the algorithm maximizes the Quantum ELBO by varying $\eta$ while keeping the model parameters fixed. In the M Step, the algorithm. In the M step, $\theta$ is updated while keeping $\eta$ fixed. We provide a detailed commentary on how this mirrors the classical EM algorithm in **lines 213 to 222**.
>
> **Typographical Suggestion: Figure 1: Why we cannot have the consistent name of the proposed method?
> Discussion: Could we use consistent notations? For example, you can keep the capitalization of the latent variable models or not. But they should be consistent.**
>
> Figure 1(a) compares two different methods. Figure 1(b) and 1(c) compare different models with the same training methodology (Contrastive Divergence on QiDBMs and DBMs). We will update Figure 1(c) legends to read QiDBM and DBM.
>
> We hope our responses have addressed the reviewer’s concerns and that the paper is now viewed more favorably.

---

> > ### Comment · Reviewer_fThS · 2025-08-04
> >
> > Reading through the other reviewers' comments, as well as the author's latest empirical experiments, I decided to keep my score for the following two concerns.
> >
> > (1) The paper presents abundant theorems without clear motivations and enough clarity. Though I appreciate the authors' efforts in trying to provide some explanations in the rebuttal, I am not sure if the full paper would be accessible to the majority of the audience or not. And it is very hard to evaluate its soundness in cases of overuse of notations.
> >
> > (2) The empirical performance remains a big challenge. Besides the unclear theoretical statements, the empirical performance, especially the newly added ones on Celeba, makes it really skeptical regarding the usefulness of the proposed method. I am just unsure how to make conclusions from baselines of having hundreds FIDs.
> >
> > Therefore, I decided to keep my score.

---

> ### Author Response · Authors · 2025-08-04
>
> We request the reviewer to go through the following points:
>
> *(1) The paper presents abundant theorems without clear motivations and enough clarity. Though I appreciate the authors' efforts in trying to provide some explanations in the rebuttal, I am not sure if the full paper would be accessible to the majority of the audience or not. And it is very hard to evaluate its soundness in cases of overuse of notations.*
>
> We present more motivation in the hope of further clarifying our results. Density Operators were first introduced to NeurIPS by Manfred Warmuth at **NeurIPS 2005** [1]. Unfortunately, due to operator theoretic difficulties, he and his co-authors concluded that *"at this point we have no convincing application for the new probability calculus"* (page 38, paragraph 2 in [2]). The approaches then could not yield any algorithms which can scale to any meaningful real-world data.
>
> With the rapid progress in quantum computing, it's crucial for the machine learning community to take the initiative in exploring how generative AI can evolve within quantum settings. We frame this challenge as the task of developing algorithms capable of learning Density Operator-based Latent Variable Models (DOLVMs) from data. Specifically, we ask: **Can Quantum Boltzmann Machines (QBMs), a key example of DOLVMs, hold their own against Deep Boltzmann Machines (DBMs)?**
>
> The current state of QBMs is limited, **supporting only up to 12 logical qubits**. In contrast, applying them to datasets on the scale of MNIST would demand roughly 6272 logical qubits, highlighting a **major scalability gap** that must be addressed.
> The EM algorithm has had a significant impact on modern generative AI, particularly in enabling models to effectively handle real-world data. In a similar spirit, our work aims to extend these capabilities. Our results at scale are driven by the integration of three key components: the **DO-EM algorithm (Algorithm 1), the decomposition of Density Operator models into classical subspaces (Theorem 5.1), and Contrastive Divergence**. Both DO-EM and CD play central roles in our empirical evaluations involving Quantum interleaved Deep Boltzmann Machines (QiDBMs) and Quantum Gaussian Restricted Boltzmann Machines (QGRBMs).
>
> We have taken great care to provide rigorous justifications and formal proofs in both the main text and appendix. To address challenges previously encountered in works such as Warmuth [1–2], we leverage **well-established concepts from quantum information theory**, including the Petz Recovery Map. While these formalisms are relatively new to the machine learning community, **we believe NeurIPS, as a flagship ML conference, is well-positioned to champion the adoption of novel theoretical tools that advance machine learning in the context of emerging quantum technologies**. We are happy to provide additional commentary or clarification to further support our theoretical results.
>
> *(2) The empirical performance remains a big challenge. Besides the unclear theoretical statements, the empirical performance, especially the newly added ones on Celeba, makes it really skeptical regarding the usefulness of the proposed method. I am just unsure how to make conclusions from baselines of having hundreds FIDs.*
>
> We now address the empirical performance of QBMs. As noted in the previous paragraph, our goal is **not to achieve state-of-the-art performance** on MNIST or CelebA-32, but rather to **demonstrate that Quantum Boltzmann Machines (QBMs) can be made competitive with Deep Boltzmann Machines (DBMs) on these benchmarks**. While it is well understood that DBMs are not the top-performing generative models, they serve as a **useful testbed for developing and evaluating new learning algorithms**. Our results show that **EM-style methods can bring Density Operator models to a level of performance comparable to classical probabilistic models**. This opens up promising directions for synergy between Quantum Computing and Generative AI.
>
> [1] Warmuth, M. K. (2005). A Bayes rule for density matrices. Advances in Neural Information Processing Systems, 18.
> [2] Warmuth, M. K., & Kuzmin, D. (2010). Bayesian generalized probability calculus for density matrices. Machine Learning, 78, 63–101.

---

> ### Comment · Reviewer_fThS · 2025-08-05
>
> I thank you for your efforts for further clarification. I understand the challenges of implementing quantum algorithms in practice. And I appreciate the efforts. I fully agree with the emphasis of the abstract on your goal of delivering a practical generative model using quantum computing; however, I am still concerned that your latest results fall short of your claim. For instance:
>
>  > We observe that the quantum variant achieves 40-50% lower FID than a GRBM of the same hidden dimension for the same amount of training time. After 250 epochs, the GRBM achieves an FID of 245.77, and the QGRBM achieves an FID of 151.33, a ~40% decrease. After 600 epochs, the GRBM achieves an FID of 134.71 compared to the QGRBM, which achieves an FID of 59.83, a ~50% decrease. We will make all code available as allowed per conference guidelines.
>
> A FID of 59.83 would typically indicate very poorly generated images, and I doubt it would indicate something affirmative. When you train it longer, would you be able to see further improvements?

---

> ### Author Response · Authors · 2025-08-05
>
> We thank the reviewer for engaging with our response and providing new comments. We request the reviewer to go through the following points:
>
> 1. **your goal of delivering a practical generative model using quantum computing**
>
> As quantum machine learning remains in its infancy, our objective is not to deliver a practical quantum generative model, but to establish a strong foundational benchmark. **Just as Deep Boltzmann Machines served as a standard testbed in classical generative modeling, our aim is to bring Quantum Boltzmann Machines to a comparable level of capability.** To this end, we present the first samples of MNIST and CelebA-32 scale data generated using Density Operator Models.
>
> While the generated images do not yet match the quality of modern deep neural network-based models, **our results clearly outperform those of classical DBMs and GRBMs**. We believe that continued research into Density Operator Models can yield practical generative models that retain this advantage over their classical counterparts. **Our work lays the algorithmic groundwork for advancing this direction.**
>
> 2. **When you train it longer, would you be able to see further improvements?**
>
> We believe that the FID values provided by the QGRBM can be significantly improved. As evidence,
> - **Not hyperparameter tuned:** Due to the time constraints of the rebuttal period, we are unable to perform any hyperparameter tuning on the QGRBM used to train CelebA-32. We use the hyperparameters provided by [1] as a good first guess. However, as the model is fundamentally different with Quantum bias terms in the hidden units, we can expect better performance after the model is properly tuned.
> - **Only run for 600 epochs:** The current SoTA paper on GRBMs [1] recommends a training time of 10,000 epochs for CelebA-32. Due to time constraints in the rebuttal process, we have only been able to run 600 epochs. As we notice a steady drop in FID values across epochs, we believe that this trend will continue. We provide a table indicating the trend in FID values across epochs on the CelebA-32 dataset.
>
> | Epoch | GRBM FID | QGRBM FID | Relative Decrease |
> |:-------:|:----------:|:-----------:|:--------------------:|
> | 20    | 363.62   | 320.17    | **11.9%**          |
> | 250   | 245.77   | 151.33    | **38.4%**         |
> | 600   | 134.71   | 59.83     | **55.6%**         |
>
> While our present experiments already provide a clear trend of improvement, we believe that samples with Quantum GRBMs can be improved significantly with appropriate hyperparameter tuning and longer train times.
>
> [1] Liao, R., Kornblith, S., Ren, M., Fleet, D. J., & Hinton, G. (2022). Gaussian-Bernoulli RBMs Without Tears. arXiv:2210.10318.

---

> > ### Comment · Reviewer_fThS · 2025-08-06
> >
> > It is not guaranteed that the FIDs would decay as much as the trend you provided in the earlier epochs. I still think a complete comparison with GRBM is necessary to fully evalutate your performance.

---

> > > ### Author Response · Authors · 2025-08-08
> > >
> > > With under 24 hours left in the discussion period, we present our most recent CelebA-32 experimental results.
> > >
> > > | Epoch | GRBM FID | QGRBM FID | Relative Decrease |
> > > |:-----:|:--------:|:---------:|:-----------------:|
> > > | 600 | 134.71	| 59.83 | **55.6%** |
> > > |  650  |  88.30   |   55.54   |    **37.1%**      |
> > > |  700  | 119.60   |   57.28   |    **52.1%**      |
> > > |  750  |  98.99   |   54.65   |    **44.8%**      |
> > > |  800  |  81.36   |   61.13   |    **24.9%**      |
> > > |  850  | 112.08   |   54.35   |    **51.5%**      |
> > > |  900  |  77.38   |   53.82   |   **30.5%**      |
> > >
> > > **This reflects the same FID trend for GRBMs and QGRBMs on the CelebA-32 dataset as observed for DBMs and QiDBMs on MNIST in Fig. 1(b) of our manuscript**. While the relative decrease fluctuates similar to Fig. 1(b), we observe that the **quantum variants studied here outperform their classical counterparts across models and datasets**. We shall include a detailed comparison of GRBMs and QGRBMs in our final draft if the paper is accepted.
> > >
> > > **Regarding high FID scores:** As is standard practice, we provide qualitative MNIST samples to supplement quantitative FID scores in **Appendix D**. Unfortunately, conference policy prevents us from sharing similar CelebA samples during the discussion period.
> > >
> > > We hope the additional experimental validation addresses the reviewer’s concerns and leads to a more favorable evaluation of our paper.

---

### Official Review · Reviewer_BprH · 2025-07-03

**Clarity:** 2
**Significance:** 2
**Originality:** 3
**Rating:** 3
**Confidence:** 2

**Summary:**

The manuscript introduces a novel Density Operator Expectation-Maximization (DO-EM) algorithm aimed at training Density Operator Latent Variable Models (DO-LVMs) on classical data using quantum-inspired methods. The core contribution lies in developing an Expectation-Maximization framework based on Quantum Information Projection (QIP), overcoming challenges in classical EM methods by addressing the absence of well-defined quantum conditional probabilities. The authors present an iterative procedure that ensures non-decreasing log-likelihood across iterations, and demonstrate the practical application of their method using Quantum Interleaved Deep Boltzmann Machines (QiDBMs), a quantum analog of classical Deep Boltzmann Machines (DBMs). Empirical results on the MNIST dataset show that QiDBMs outperform classical DBMs in terms of Fréchet Inception Distance (FID).

**Questions:**

The questions are included in the weakness.

**Ethical Concerns:**

["NO or VERY MINOR ethics concerns only"]

**Limitations:**

The limitations are included in the weakness.

**Paper Formatting Concerns:**

No formatting concerns.

**Quality:**

3

**Strengths And Weaknesses:**

Strengths:
- The paper develops a quantum-inspired Expectation-Maximization algorithm that scales to real-world data. The introduction of Quantum Information Projection is a novel and effective approach.
- The paper provides a rigorous theoretical analysis, contributing to a solid foundation for the proposed algorithm.

Weakness:
- The manuscript presents several theorems and lemmas related to the DO-EM algorithm, but these are not sufficiently explained or contextualized. As a reader with limited expertise in this field, it is difficult to fully grasp the purpose and implications of these theoretical results.
- The experimental evaluation is somewhat limited. Currently, the authors only provide results on the MNIST dataset, which may not be enough to demonstrate the general effectiveness of the proposed approach across a variety of tasks and datasets.

---

> ### Author Rebuttal · Authors · 2025-07-31
>
> We thank the reviewer for their review. We are pleased to note that the reviewer finds the DO-EM algorithm and the Quantum Information Projection novel and effective. We answer specific questions raised by the reviewer below:
>
> **The manuscript presents several theorems and lemmas related to the DO-EM algorithm, but these are not sufficiently explained or contextualized. As a reader with limited expertise in this field, it is difficult to fully grasp the purpose and implications of these theoretical results.**
>
> Existing Generative models developed in the context of Quantum computing **rarely scale beyond 10 qubits**. The purpose of developing an EM style Algorithm is to create a generic procedure that can be applied to Latent Variable models described through Density Operators (DOLVMs). There is growing interest in the Quantum Computing community in such models, as they can provide principled alternatives to existing circuit-based models. The DOEM Algorithm is developed for learning DOLVM as a parallel to the well-known EM algorithm for Latent Variable Models.
> Standard theoretical techniques, like Jensen’s Inequality, cannot be applied in the settings considered here to analyze the behaviour of density operators. In our work, we show that the models we are proposing **provably work** with DOLVM. This requires several new theoretical insights and results. We highlight the following:
>
> Lemma 4.1 (Quantum ELBO): As discussed in lines 146-154, the log-likelihood of a density operator latent variable model is not easy to maximize due to non-commuting (i.e, AB-BA!=0) operators and the partial trace operation. In classical machine learning, similar issues are solved using the Evidence Lower Bound technique (EM algorithm [1], Variational Autoencoders [2], Diffusion [3], etc.). The Quantum Evidence Lower Bound is a first step in addressing these problems for Density Operators. It is significantly easier to maximize the QELBO using an appropriate $\eta$.
>
> Theorem 4.3: We solve an instance of a Quantum Information Projection problem and tie it to the well-known Petz Recovery Map. This result is of independent interest since it ties the Quantum I-Projection problem to well-known results in Quantum Information Theory. Most importantly, it allows us to generalize the classical E-step using an information geometric interpretation.
>
> Theorem 4.4 + Corollary 4.5: Commentary provided in lines 211-222.
>
> Corollary 4.6: This result shows us that the DO-EM algorithm makes training Hamiltonian-based models significantly easier by removing the partial trace operation. This mirrors the utility of the EM algorithm for classical latent variable models.
>
> Theorem 5.1 + Lemma 5.2: Our results in Section 4 were for both quantum and classical data. Theorem 5.1 proves that for datasets without quantum correlations, the DO-LVMs that satisfy Condition S are models that have no quantum correlations on the visible variables (CQ-LVMs). The density operators of these models are decomposable into smaller subspaces, making it easier to store and train. Lemma 5.2 uses this decomposition to simplify the log-likelihood expression for DO-LVMs. This simplified expression allows the calculation of log-likelihood for each datapoint independently (since the visible states do not have quantum correlations). This allows training in batches - something that is ubiquitous in machine learning.
> Lemma 5.4 + Algorithm 2: These results specialize Algorithm 1 to CQ-LVMs - making it computationally and memory efficient while also allowing batching of datasets during training.
>
> Corollary 5.3 + Corollary 5.5: Specializes Theorem 5.1 to Hamiltonian-based models and then to Quantum Boltzmann Machines, the most prominent DO-LVMs in literature. These results show that the structure imposed by Theorem 5.1 is not restrictive. We show in Fig 1(a) that these models do as well as existing QBMs in a faster time frame.
>
>
> **The experimental evaluation is somewhat limited. Currently, the authors only provide results on the MNIST dataset, which may not be enough to demonstrate the general effectiveness of the proposed approach across a variety of tasks and datasets.**
>
>
> We wish to bring to the reviewer’s notice that prior work in the area of applying quantum algorithms rarely scaled beyond 12 qubits of data. In this regard, even a dataset such as MNIST was far beyond the reach of prior art. Our work shows the scalability to 6000+ qubits as indicated in our experiments.
> To further demonstrate the validity of our work, we provide preliminary experiments on a downscaled version of CelebA containing 32x32-sized images, which contains ~200,000  images of celebrity faces. We train a Gaussian RBM [A] and compare it with our proposed algorithm. We observe that after 20 epochs of training, the Gaussian RBM achieves an FID of 363.622 measured against the CelebA test set. With the same hyperparameter setting, the quantum variant as described in this work achieves an FID of 320.17, **a 10% relative improvement**. We note that we did not tune the hyperparameters of the Quantum variant due to the time taken for these experiments. Our results indicate that our proposed algorithm for **quantum-enhanced RBMs can outperform traditional approaches with RBMs**. Our results can be further expanded and enhanced in this setting.
>
> We hope that the theoretical strength of this work, combined with the additional empirical results presented in this review, clarifies the concerns raised by the reviewer. The reviewer’s comments have greatly enhanced our manuscript by helping contextualize the theoretical results presented in our work. We believe that the additional empirical experiments have also strengthened our work.
>
> **We invite the reviewer to participate in the discussion period to discuss any further clarifications preventing the reviewer from increasing their scores.**
>
> [A] Gaussian RBMs without tears, https://arxiv.org/abs/2210.10318

---

> > ### Comment · Reviewer_BprH · 2025-08-06
> >
> > I appreciate the authors' response. However, as noted by Reviewer fThS, the manuscript contains numerous theorems that lack clear motivations, sufficient clarity, and an excessive use of notations. Despite the additional clarifications provided in the subsequent response, it is still hard for me to evaluate the soundness of this paper based on the current version of the manuscript. Therefore, I will maintain my original score.

---

> ### Comment · Area_Chair_oePb · 2025-08-04
>
> Dear Reviewer BprH,
>
> The Author–Reviewer discussion period has begun. Please read the rebuttal and let us know whether it satisfactorily addresses your concerns. If not, could you specify what remains inadequate? Your response will help us evaluate the paper and assist the authors in improving their work.
>
> Thanks!
> Best,
> AC

---

> > ### Comment · Area_Chair_oePb · 2025-08-06
> >
> > Dear Reviewer BprH,
> >
> > Please read the author rebuttal and let us know whether it addresses your concerns. Reviewer fThS and Reviewer W6WS also raised questions about the empirical verification. Their discussions with the authors may be useful for your reference.
> >
> > If you do not respond, I will need to apply the “insufficient review” flag to your review in accordance with the NeurIPS 2025 policy.
> >
> > > Reviewers must participate in discussions with authors before submitting “Mandatory Acknowledgement”. You can flag non-participating reviewers with the InsufficientReview button, and we will use the flag in accordance with possible penalties of this year's responsible reviewing initiative and future reviewing invitations.
> >
> > Best,
> > AC

---

> ### Author Response · Authors · 2025-08-07
>
> Dear reviewer BprH, thank you for the response. We request you to go through the other responses we have provided. However, for the sake of clarity we present the following points in response to your comment.
>
> > However, as noted by Reviewer fThS, the manuscript contains numerous theorems that lack clear motivations, sufficient clarity, and an excessive use of notations.
>
> **Lack of Motivation:** Density Operators are quantum analogs of probability distributions. There has been increased interest in using them as models for machine learning due to the emergence of quantum technologies. At present, there are no scalable training algorithms for Density Operator-based Latent Variable Models (DO-LVMs) applicable to real-world datasets. Existing methods are restricted to extremely low-dimensional inputs, such as bit strings of length 12, and have yet to demonstrate efficacy beyond toy problems. Inspired by the success of the Expectation-Maximization algorithm in classical Latent Variable Models and generative AI, we attempt to derive an EM-like framework for Density Operators. Our hope is that such an algorithm will help DO-LVMs scale to real-world data.
>
> The classical EM algorithm as described in line 88 requires a minorant function ($Q_i$) of the log-likelihood:
> \begin{align*}
>     \theta^{(k+1)} &= \underset{\theta}{\mathrm{argmax}} \frac1N\sum_{i=1}^N Q_i(\theta\mid\theta^{(k)}),
> \end{align*}
> where $\ell_i(\theta) \ge Q_i(\theta|\theta^{(k)})$ and $\ell_i(\theta^{(k)}) = Q_i(\theta^{(k)}|\theta^{(k)})$. By maximizing the minorant function at each $\theta^{(k)}$ to obtain a new parameter, the algorithm increases the log-likelihood. As the minorant function $Q_i$ is significantly easier to compute and maximize, the EM algorithm is a much better alternative to directly maximizing the log-likelihood by gradient ascent for latent variable models.
>
> **Clarity:** The structuring of the paper is as follows.
> In Section 3, we define Density Operator Latent Variable Models and describe the need for an EM algorithm for these models in lines 146-161. The development of such an EM algorithm requires three things. These are all open questions in the context of Density operators.
> 1. The identification of a suitable $Q_i$ function.
> 2. The algorithm guaranteeing an increase in log-likelihood
> 3. Exploration of the sufficiency condition used to obtain the guarantee and their application to models such as Quantum RBMs
>
> We answer questions 1 and 2 in Section 4. We have provided a detailed, non-mathematical commentary on each of our theorems in our rebuttal to Reviewers BprH and FuJL. We request the referee to go through those points.
>
> We answer question 3 in Section 5. In order to do that, we specialize our results in Section 4 to classical data. We show that DO-LVMs that learn on classical data have additional structure that can be exploited to make training easier. We also show that this class of models with additional structure is an interesting one with several notable examples such as the Quantum RBM and the Quantum interleaved Deep Boltzmann Machine.
>
> **Manuscript contains numerous theorems:** As described above, our theoretical results are necessary to establish the soundness and utility of our algorithm. While these results could have been split into several papers, we wish to provide one self-contained and strong paper that is backed by theoretical guarantees. We are happy to incorporate suggestions which will improve the presentation of the paper.
>
> **Excessive use of notation:** The formalisms in quantum mechanics forces us to work with Operator Theory and Quantum Information Theory to derive our theoretical results. We introduce new techniques to the machine learning community and set the foundation for further exploration of Density Operator Models. The tools we use are standard in any QIT textbook [1]. As the flagship conference in machine learning, we believe that NeurIPS should be at the forefront of adopting and adapting to new techniques as new technology emerges.
>
> [1] Wilde, Mark M. Quantum information theory. Cambridge university press, 2013.

---

### Author Response · Authors · 2025-08-08
**Summary of Experiments in the manuscript**

We provide below a concise summary of our experimental results for the reviewers’ convenience. Our study comprises three key experiments:

1. **Baseline: DO-EM vs. state-of-the-art QBM training with hidden units**

Our first experiment benchmarks the DO-EM algorithm against the gradient ascent approach in [1], the largest fully connected QBM with hidden units trained to date. As noted in [1], their method *“renders the training of a QBM inefficient and basically impractical for large systems,”* and their workaround *“cannot learn the transverse field,”* leaving quantum bias terms as fixed hyperparameters. Sections 3–5 of our paper present theoretical results that **resolve both issues**. In Fig. 1(a), we show that DO-EM matches the performance of direct gradient ascent while using lesser parameters and being significantly easier and faster to implement.

2. **DO-EM at scale: training on 6272 visible qubits**

Here, we train a Quantum interleaved Deep Boltzmann Machine (QiDBM) on binarized 8-bit MNIST (6272 visible units). This represents a ~500× scale-up over the current QBM state of the art (12 visible units). In the absence of large-scale QBM benchmarks, we compare against Deep Boltzmann Machines (DBMs) and obtain an FID of 45% for a model of similar size (Fig. 1(b)). Qualitative samples from this experiment are shown in Appendix D.

3. **Performance efficiency of DO-EM: quality vs. model size**

Finally, we train DBMs and QiDBMs on binarized MNIST while varying the number of parameters (via hidden units). As shown in Fig. 1(c), QiDBMs with fewer parameters can generate images with lower FIDs than larger classical DBMs.

Our main contribution is **an algorithmic framework for training DO-LVMs on real-world data** rather than toy datasets. This framework allows QiDBMs and QRBMs to be trained on any dataset that supports DBMs or RBMs using identical computational resources. Independently of performance metrics, **our experiments establish a ~500× scale-up in QBM training, from 12 visible qubits to 6272 visible qubits, marking the largest QBM ever trained in literature**. This scale result stands on its own. In addition, we observe that **DO-based models can outperform their classical counterparts in MNIST image generation**, highlighting the potential of Density Operator Models in future ML research.

We hope that this summary helps the reviewers appreciate the strength of our experimental contributions.

[1] Amin, M. H., Andriyash, E., Rolfe, J., Kulchytskyy, B., & Melko, R. (2018). Quantum boltzmann machine. Physical Review X, 8(2), 021050.

---

### Author Response · Authors · 2025-08-09
**Additional experimental results on Fashion-MNIST**

We provide additional experimental validation on Fashion-MNIST, offering further evidence that Density Operator models can outperform their classical counterparts. In particular, we evaluate a Gaussian Restricted Boltzmann Machine (GRBM) with 10,000 hidden units, trained for 3000 epochs following the setup in [1]. We adapt the GRBM (as in our CelebA experiments) into our proposed Quantum GRBM (QGRBM). Using the identical training setup as the GRBM, **without any additional hyperparameter tuning**, the QGRBM achieves a 20% reduction in FID. The results, along with the relative FID decrease (lower is better), are summarized in the table below.

| Epoch      | 1500              | 2000              | 2500              | 3000               |
|------------|-------------------|-------------------|-------------------|--------------------|
| GRBM FID   | 138.61            | 125.77            | 119.37            | 112.93             |
| QGRBM FID  | 115.88 (-16.4%)   | 94.01 (-25.3%)    | 101.29 (-15.2%)   | **91.99 (-18.5%)** |

We are pleased to note that the proposed method continues to outperform classical Boltzmann machines, even after extending them to Gaussian RBMs.

[1] Liao, R., Kornblith, S., Ren, M., Fleet, D. J., & Hinton, G. (2022). Gaussian-bernoulli rbms without tears. arXiv preprint arXiv:2210.10318.

---

### Author Response · Authors · 2025-08-09
**Summary note**

We thank the reviewers for their active participation in the discussion period. Your questions and feedback have helped us improve both the clarity and scope of our work. As the discussion phase concludes, we summarize our main contributions and highlight the clarifications and additional experiments provided during this process.

### Summary of Contributions

This paper presents the first principled derivation of an Expectation-Maximization framework for density-operator latent-variable models, connecting the E-step to quantum information projections and the Petz recovery map, and providing full proofs of likelihood ascent. Using Quantum Boltzmann Machines as a theoretical testbed, we achieve a scale-up from 12 to 6272 visible units (over two orders of magnitude) alongside large, consistent relative improvements in FID against matched baselines. The contribution is foundational: a provable, scalable training procedure for quantum-inspired generative models, not a claim to surpass modern classical generative models. This type of technically solid advance, expanding the scope of what is feasible, is well aligned with NeurIPS’ tradition of valuing foundational work that future applied research can build upon.

### Discussions and Additional Experiments

- **Quantum Resources**: We provide a detailed discussion of the quantum resources required to implement our proposed algorithm. We provided a potential implementation of the DO-EM algorithm on a quantum device based on existing quantum subroutines. Since such subroutines are an active area of research, we deferred a full analysis of such an implementation to future work.

-  **Additional Experiments**: We extend our approach to Gaussian RBMs, enabling application to additional real-world datasets. On FashionMNIST and CelebA, our Quantum RBMs (QRBMs) continue to outperform their classical RBM counterparts, confirming that our method provides consistent benefits across architectures and datasets.

We believe this work establishes an essential bridge between quantum-inspired theory and generative AI. By providing both rigorous guarantees and concrete, scalable algorithms, it lays a foundation that the community can quickly build upon. We hope the reviewers recognize the significance of this contribution and agree that it merits presentation at NeurIPS.

---

### Note · Authors · 2025-08-15

We thank the reviewers and AC for their comments. We summarize the **significance** of the results and the **key rebuttal points** to assist the AC.

**Significance**

With the increasing interest in quantum technology, there is a need for foundational understanding of Density Operators (DO): quantum generalizations of probability distributions. Developing DO-based models for generative AI is crucial, yet existing approaches cannot be applied to real-world data. Motivated by the success of the EM algorithm in generative AI, we develop an EM algorithm for DOs and establish its theoretical foundations. Unlike prior methods which can only handle 10–12 units (1 bit MNIST requires 700+ units), the proposed method can handle 6200+ units. This represents a significant advance in the field and, given the rise of quantum technology, should be of strong interest to the NeurIPS community.

**Important Concerns and Rebuttals**

1. Suitability at NeurIPS (fThS)
As the leading ML venue, it is well-suited for foundational results at the intersection of quantum technology and ML. DOs were introduced to ML by **M. Warmuth at NeurIPS ‘05**, where applying EM to real-world data was identified as an open question. Our work resolves this long-standing problem and advances the use of DO models for generative modeling.
2. Too many theoretical results (fThS, Bprh)
Developing the DO-EM algorithm and showing its applicability required new formalism, with many steps paralleling standard EM. We present only the essential results, though they can be extended in many directions. Reviewers noted the **novelty of the ideas** (FuJL) and **found the proofs and derivations technically sound** (W6WS).
3. Comparing DBM vs QiDBM on more datasets (W6WS)
We provided new experiments on Gaussian QRBMs on two more datasets, F-MNIST and CelebA-32, confirming trends observed in Fig. 1(b). While the experiments in the manuscript validated all claims in the paper (correctness, scalability, and enhanced performance *vs DBMs*), these experiments provide additional validation to our claims.
4. Needs quantum resource analysis (FuJL)
We have provided a detailed resource analysis for an end-to-end quantum implementation of DO-EM using existing subroutines, highlighting its applicability to classical data.

If accepted, the paper will be revised to include clarifying remarks made during the rebuttal, the additional experiments on new models and datasets, and the resource analysis on quantum devices.

---

### Decision · Program_Chairs · 2025-09-17

**Decision:**

Reject

**Comment:**

This paper proposes a generalization of the EM algorithm using the notion of density operators from quantum information theory. The proposed algorithm and analysis are novel. Nevertheless, this paper fails to provide compelling empirical results, as pointed out by Reviewers fThS and W6WS, and its presentation is not very accessible to the machine learning community, as noted by Reviewers BprH and fThS. I agree with both points and therefore suggest rejecting this paper.

Some additional comments: The discussion about the need for an EM algorithm in Section 3, as well as the literature review in Appendix B.1, should be moved to the introduction. Doing so helps clarify the motivation, novelty, and significance of this work. Moreover, it would be beneficial to provide a clearer comparison with existing work in terms of the ideas, in addition to the results.